# SELF-STABILIZATION: THE IMPLICIT BIAS OF GRADIENT DESCENT AT THE EDGE OF STABILITY

**Alex Damian\*, Eshaan Nichani\* & Jason D. Lee**
Princeton University
{ad27,eshnich,jasonlee}@princeton.edu

## ABSTRACT

Traditional analyses of gradient descent show that when the largest eigenvalue of the Hessian, also known as the sharpness $S(\theta)$, is bounded by $2/\eta$, training is "stable" and the training loss decreases monotonically. Recent works, however, have observed that this assumption does not hold when training modern neural networks with full batch or large batch gradient descent. Most recently, Cohen et al. (2021) detailed two important phenomena. The first, dubbed *progressive sharpening*, is that the sharpness steadily increases throughout training until it reaches the instability cutoff $2/\eta$. The second, dubbed *edge of stability*, is that the sharpness hovers at $2/\eta$ for the remainder of training while the loss continues decreasing, albeit non-monotonically. We demonstrate that, far from being chaotic, the dynamics of gradient descent at the edge of stability can be captured by a cubic Taylor expansion: as the iterates diverge in direction of the top eigenvector of the Hessian due to instability, the cubic term in the local Taylor expansion of the loss function causes the curvature to decrease until stability is restored. This property, which we call *self-stabilization*, is a general property of gradient descent and explains its behavior at the edge of stability. A key consequence of self-stabilization is that gradient descent at the edge of stability implicitly follows *projected* gradient descent (PGD) under the constraint $S(\theta) \leq 2/\eta$. Our analysis provides precise predictions for the loss, sharpness, and deviation from the PGD trajectory throughout training, which we verify both empirically in a number of standard settings and theoretically under mild conditions. Our analysis uncovers the mechanism for gradient descent's implicit bias towards stability.

## 1 INTRODUCTION

### 1.1 GRADIENT DESCENT AT THE EDGE OF STABILITY

Almost all neural networks are trained using a variant of gradient descent, most commonly stochastic gradient descent (SGD) or ADAM (Kingma & Ba, 2015). When deciding on an initial learning rate, many practitioners rely on intuition drawn from classical optimization. In particular, the following classical lemma, known as the "descent lemma," provides a common heuristic for choosing a learning rate in terms of the sharpness of the loss function:

**Definition 1.** *Given a loss function $L(\theta)$, the sharpness at $\theta$ is defined to be $S(\theta) := \lambda_{max}(\nabla^2 L(\theta))$. When this eigenvalue is unique, the associated eigenvector is denoted by $u(\theta)$.*

**Lemma 1** (Descent Lemma). *Assume that $S(\theta) \leq \ell$ for all $\theta$. If $\theta_{t+1} = \theta_t - \eta\nabla L(\theta_t)$,*

$$L(\theta_{t+1}) \leq L(\theta_t) - \frac{\eta(2 - \eta\ell)}{2}\|\nabla L(\theta_t)\|^2.$$

Here, the loss decrease is proportional to the squared gradient, and is controlled by the quadratic $\eta(2 - \eta\ell)$ in $\eta$. This function is maximized at $\eta = 1/\ell$, a popular learning rate criterion. For any $\eta < 2/\ell$, the descent lemma guarantees that the loss will decrease. As a result, learning rates below $2/\ell$ are considered "stable" while those above $2/\ell$ are considered "unstable." For quadratic

---

*Equal contribution

loss functions, e.g. from linear regression, this is tight. Any learning rate above $2/\ell$ provably leads to exponentially increasing loss.

However, it has recently been observed that in neural networks, the descent lemma is not predictive of the optimization dynamics. Recently, Cohen et al. (2021) observed two important phenomena for gradient descent, which made more precise similar observations in Jastrzębski et al. (2019); Jastrzebski et al. (2020) for SGD:

**Progressive Sharpening** Throughout most of the optimization trajectory, the gradient of the loss is negatively aligned with the gradient of sharpness, i.e. $\nabla L(\theta) \cdot \nabla S(\theta) < 0$. As a result, for any reasonable learning rate $\eta$, the sharpness increases throughout training until it reaches $S(\theta) = 2/\eta$.

**Edge of Stability** Once the sharpness reaches $2/\eta$ (the "break-even" point in Jastrzebski et al. (2020)), it ceases to increase and remains around $2/\eta$ for the rest of training. The descent lemma no longer guarantees the loss decreases but the loss still continues decreasing, albeit non-monotonically.

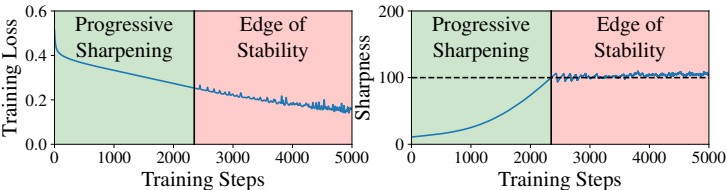

## 1.2 SELF-STABILIZATION: THE IMPLICIT BIAS OF INSTABILITY

In this work we explain the second stage, "edge of stability." We identify a new implicit bias of gradient descent which we call *self-stabilization*. Self-stabilization is the mechanism by which the sharpness remains bounded around $2/\eta$, despite the continued force of progressive sharpening, and by which the gradient descent dynamics do not diverge, despite instability. Unlike progressive sharpening, which is only true for specific loss functions (e.g. those resulting from neural network optimization (Cohen et al., 2021)), self stabilization is a general property of gradient descent.

Traditional non-convex optimization analyses involve Taylor expanding the loss function to second order around $\theta$ to prove loss decrease when $\eta \leq 2/S(\theta)$. When this is violated, the iterates diverge exponentially in the top eigenvector direction, $u$, thus leaving the region in which the loss function is locally quadratic. Understanding the dynamics thus necessitates a *cubic* Taylor expansion.

Our key insight is that the missing term in the Taylor expansion of the gradient after diverging in the $u$ direction is $\nabla^3 L(\theta)(u, u)$, which is conveniently equal to the gradient of the sharpness at $\theta$:

**Lemma 2** (Self-Stabilization Property). *If the top eigenvalue of $\nabla^2 L(\theta)$ is unique, then the sharpness $S(\theta)$ is differentiable at $\theta$ and $\nabla S(\theta) = \nabla^3 L(\theta)(u(\theta), u(\theta))$.*

As the iterates move in the negative gradient direction, this term has the effect of *decreasing the sharpness*. The story of self-stabilization is thus that as the iterates diverge in the $u$ direction, the strength of this movement in the $-\nabla S(\theta)$ direction grows until it forces the sharpness below $2/\eta$, at which point the iterates in the $u$ direction shrink and the dynamics re-enter the quadratic regime.

This negative feedback loop prevents both the sharpness $S(\theta)$ and the movement in the top eigenvector direction, $u$, from growing out of control. As a consequence, we show that gradient descent *implicitly* solves the *constrained minimization problem*:

$$\min_{\theta} L(\theta) \quad \text{such that} \quad S(\theta) \leq 2/\eta. \tag{1}$$

Specifically, if the stable set $\mathcal{M}$ is defined by $\mathcal{M} := \{\theta \ : \ S(\theta) \leq 2/\eta \text{ and } \nabla L(\theta) \cdot u(\theta) = 0\}$[1] then the gradient descent trajectory $\{\theta_t\}$ tracks the following projected gradient descent trajectory $\{\theta_t^\dagger\}$ which solves the constrained problem (Barber & Ha, 2017):

$$\theta_{t+1}^\dagger = \text{proj}_{\mathcal{M}}\left(\theta_t^\dagger - \eta \nabla L(\theta_t^\dagger)\right) \quad \text{where} \quad \text{proj}_{\mathcal{M}}(\theta) := \operatorname*{arg\,min}_{\theta' \in \mathcal{M}} \|\theta - \theta'\|. \tag{2}$$

---

[1]The condition that $\nabla L(\theta) \cdot u(\theta) = 0$ is necessary to ensure the stability of the constrained trajectory.

Our main contributions are as follows. First, we explain self-stabilization as a generic property of gradient descent for a large class of loss functions, and provide precise predictions for the loss, sharpness, and deviation from the constrained trajectory $\{\theta_t^\dagger\}$ throughout training (Section 4). Next, we prove that under mild conditions on the loss function (which we verify empirically for standard architectures and datasets), our predictions track the true gradient descent dynamics up to higher order error terms (Section 5). Finally, we verify our predictions by replicating the experiments in Cohen et al. (2021) and show that they model the true gradient descent dynamics (Section 6).

## 2 RELATED WORK

Xing et al. (2018) observed that for some neural networks trained by full-batch gradient descent, the loss is not monotonically decreasing. Wu et al. (2018) remarked that gradient descent cannot converge to minima where the sharpness exceeds $2/\eta$ but did not give a mechanism for avoiding such minima. Lewkowycz et al. (2020) observed that when the initial sharpness is larger than $2/\eta$, gradient descent "catapults" into a stable region and eventually converges. Jastrzębski et al. (2019) studied the sharpness along stochastic gradient descent trajectories and observed an initial increase (i.e. progressive sharpening) followed by a peak and eventual decrease. They also observed interesting relationships between the dynamics in the top eigenvector direction and the sharpness. Jastrzebski et al. (2020) conjectured a general characterization of stochastic gradient descent dynamics asserting that the sharpness tends to grow but cannot exceed a stability criterion given by their eq (1), which reduces to $S(\theta) \leq 2/\eta$ in the case of full batch training. Cohen et al. (2021) demonstrated that for the special case of (full batch) gradient descent training, the optimization dynamics exhibit a simple characterization. First, the sharpness rises until it reaches $S(\theta) = 2/\eta$ at which point the dynamics transition into an "edge of stability" (EOS) regime where the sharpness oscillates around $2/\eta$ and the loss continues to decrease, albeit non-monotonically.

Recent works have sought to provide theoretical analyses for the EOS phenomenon. Ma et al. (2022) analyzes EOS when the loss satisfies a "subquadratic growth" assumption. Ahn et al. (2022) argues that unstable convergence is possible when there exists a "forward invariant subset" near the set of minimizers. Arora et al. (2022) analyzes progressive sharpening and the EOS phenomenon for normalized gradient descent close to the manifold of global minimizers. Lyu et al. (2022) uses the EOS phenomenon to analyze the effect of normalization layers on sharpness for scale-invariant loss functions. Chen & Bruna (2022) show global convergence despite instability for certain 2D toy problems and in a 1-neuron student-teacher setting. The concurrent work Li et al. (2022b) proves progressive sharpening for a two-layer network and analyzes the EOS dynamics through four stages similar to ours using the norm of the output layer as a proxy for sharpness.

Beyond the EOS phenomenon itself, prior work has also shown that SGD with large step size or small batch size will lead to a decrease in sharpness (Keskar et al., 2017; Jastrzebski et al., 2017; Jastrzębski et al., 2019; Jastrzebski et al., 2020). Gilmer et al. (2021) also describes connections between EOS, learning rate warm-up, and gradient clipping.

At a high level, our proof relies on the idea that oscillations in an unstable direction prescribed by the quadratic approximation of the loss cause a longer term effect arising from the third-order Taylor expansion of the dynamics. This overall idea has also been used to analyze the implicit regularization of SGD (Blanc et al., 2020; Damian et al., 2021; Li et al., 2022a). In those settings, oscillations come from the stochasticity, while in our setting the oscillations stem from instability.

## 3 SETUP

We denote the loss function by $L \in C^3(\mathbb{R}^d)$. Let $\theta \in \mathbb{R}^d$ follow gradient descent with learning rate $\eta$, i.e. $\theta_{t+1} := \theta_t - \eta \nabla L(\theta_t)$. Recall that $\mathcal{M} := \{\theta : S(\theta) \leq 2/\eta \text{ and } \nabla L(\theta) \cdot u(\theta) = 0\}$ is the set of stable points and $\text{proj}_{\mathcal{M}} := \arg\min_{\theta' \in \mathcal{M}} \|\theta - \theta'\|$ is the orthogonal projection onto $\mathcal{M}$. For notational simplicity, we will shift time so that $\theta_0$ is the first point such that $S(\text{proj}_{\mathcal{M}}(\theta_0)) = 2/\eta$. The constrained trajectory $\theta^\dagger$ is initialized with $\theta_0^\dagger := \text{proj}_{\mathcal{M}}(\theta_0)$ after which it follows eq. (2).

Our key assumption is the existence of progressive sharpening along the constrained trajectory, which is captured by the progressive sharpening coefficient $\alpha(\theta) := -\nabla L(\theta) \cdot \nabla S(\theta)$:

**Assumption 1** (Progressive Sharpening). *Let $\alpha(\theta) := -\nabla L(\theta) \cdot \nabla S(\theta)$. Then $\alpha(\theta_t^\dagger) > 0$.*

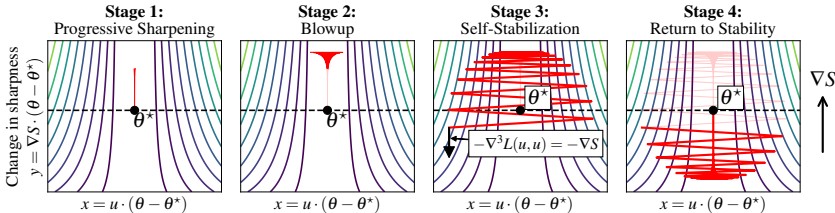

Figure 1: The four stages of edge of stability, demonstrated on a toy loss function (Appendix B).

We focus on the regime in which there is a single unstable eigenvalue, and we leave understanding multiple unstable eigenvalues to future work. We thus make the following assumption on $\nabla^2 L(\theta_t^\dagger)$:

**Assumption 2** (Eigengap). *For some absolute constant $c < 2$ we have $\lambda_2(\nabla^2 L(\theta_t^\dagger)) < c/\eta$.*

## 4 THE SELF-STABILIZATION PROPERTY OF GRADIENT DESCENT

In this section, we derive a set of equations that predict the displacement between the gradient descent trajectory $\{\theta_t\}$ and the constrained trajectory $\{\theta_t^\dagger\}$. Viewed as a dynamical system, these equations give rise to a negative feedback loop, which prevents both the sharpness and the displacement in the unstable direction from diverging. These equations also allow us to predict the values of the sharpness and the loss throughout the gradient descent trajectory.

### 4.1 THE FOUR STAGES OF EDGE OF STABILITY: A HEURISTIC DERIVATION

The analysis in this section proceeds by a cubic Taylor expansion around a fixed reference point $\theta^\star := \theta_0^\dagger$.[2] For notational simplicity, we will define the following quantities at $\theta^\star$:

$$\nabla L := \nabla L(\theta^\star), \ \nabla^2 L := \nabla^2 L(\theta^\star), \ u := u(\theta^\star), \ \nabla S := \nabla S(\theta^\star), \ \alpha := \alpha(\theta^\star), \ \beta := \|\nabla S\|^2,$$

where $\alpha = -\nabla L \cdot \nabla S > 0$ is the progressive sharpening coefficient at $\theta^\star$. For simplicity, in Section 4 we assume that $\nabla S \perp u$ and $\nabla L, \nabla S \in \ker(\nabla^2 L)$, and ignore higher order error terms.[3] Our main argument in Section 5 does not require these assumptions and tracks all error terms explicitly.

We want to track the movement in the unstable direction $u$ and the direction of changing sharpness $\nabla S$, and thus define $x_t := u \cdot (\theta_t - \theta^\star)$ and $y_t := \nabla S \cdot (\theta_t - \theta^\star)$. Note that $y_t$ is approximately equal to the change in sharpness from $\theta^\star$ to $\theta_t$, since Taylor expanding the sharpness yields

$$S(\theta_t) \approx S(\theta^\star) + \nabla S \cdot (\theta_t - \theta^\star) = 2/\eta + y_t.$$

At a high level, the mechanism for edge of stability can be described in 4 stages (see Figure 1):

**Stage 1: Progressive Sharpening** While $x, y$ are small, $\nabla L(\theta_t) \approx \nabla L$. In addition, because $\nabla L \cdot \nabla S < 0$, gradient descent naturally increases the sharpness at every step. In particular,

$$y_{t+1} - y_t = \nabla S \cdot (\theta_{t+1} - \theta_t) \approx -\eta \nabla L \cdot \nabla S = \eta \alpha.$$

The sharpness therefore increases linearly with rate $\eta \alpha$.

**Stage 2: Blowup** As $x_t$ measures the deviation from $\theta^\star$ in the $u$ direction, the dynamics of $x_t$ can be modeled by gradient descent on a quadratic with sharpness $S(\theta_t) \approx 2/\eta + y_t$. In particular, the rule for gradient descent on a quadratic gives[4]

$$x_{t+1} = x_t - \eta u \cdot \nabla L(\theta_t) \approx x_t - \eta S(\theta_t) x_t \approx x_t - \eta[2/\eta + y_t] x_t = -(1 + \eta y_t) x_t.$$

When the sharpness exceeds $2/\eta$, i.e. when $y_t > 0$, $|x_t|$ begins to grow exponentially.

---

[2]Beginning in Section 5, the reference points for our Taylor expansions change at every step to minimize errors. However, fixing the reference point in this section simplifies the analysis, better illustrates the negative feedback loop, and motivates the definition of the constrained trajectory.

[3]We give an explicit example of a loss function satisfying these assumptions in Appendix B.

[4]A rigorous derivation of this update in terms of $S(\theta_t)$ instad of $S(\theta^\star)$ requires a third-order Taylor expansion around $\theta^\star$; see Appendix I for more details.

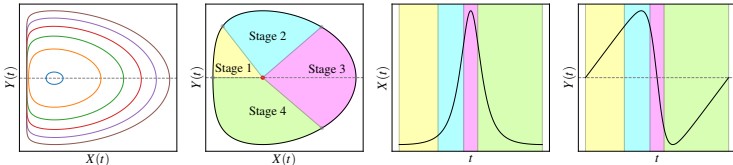

Figure 2: **The effect of $X(0)$ (left):** We plot the evolution of the ODE in eq. (4) with $\alpha = \beta = 1$ for varying $X(0)$. Observe that smaller $X(0)$'s correspond to larger curves. **The four stages of edge of stability (right):** We show how the four stages of edge of stability described in Section 4.1 and Figure 1 correspond to different parts of the curve generated by the ODE in eq. (4).

**Stage 3: Self-Stabilization**  Once the movement in the $u$ direction is sufficiently large, the loss is no longer locally quadratic. Understanding the dynamics necessitates a third order Taylor expansion. The missing cubic term in the Taylor expansion of $\nabla L(\theta_t)$ is $\nabla^3 L(u, u) \frac{x_t^2}{2} = \nabla S \frac{x_t^2}{2}$ by Lemma 2. This biases the optimization trajectory in the $-\nabla S$ direction, which decreases sharpness. Recalling $\beta = \|\nabla S\|^2$, the new update for $y$ becomes:

$$y_{t+1} - y_t = \eta \alpha + \nabla S \cdot \left( -\eta \nabla^3 L(u, u) \frac{x_t^2}{2} \right) = \eta \left( \alpha - \beta \frac{x_t^2}{2} \right)$$

Therefore once $x_t > \sqrt{2\alpha/\beta}$, the sharpness begins to decrease and continues to do so until the sharpness goes below $2/\eta$ and the dynamics return to stability.

**Stage 4: Return to Stability**  At this point $|x_t|$ is still large from stages 1 and 2. However, the self-stabilization of stage 3 eventually drives the sharpness below $2/\eta$ so that $y_t < 0$. Because the rule for gradient descent on a quadratic with sharpness $S(\theta_t) = 2/\eta + y_t < 2/\eta$ is $x_{t+1} \approx -(1 + \eta y_t) x_t$, $|x_t|$ begins to shrink exponentially and the process returns to stage 1.

Combining the update for $x_t, y_t$ in all four stages, we obtain the following simplified dynamics:

$$x_{t+1} \approx -(1 + \eta y_t) x_t \quad \text{and} \quad y_{t+1} \approx y_t + \eta \left( \alpha - \beta \frac{x_t^2}{2} \right) \tag{3}$$

where we recall $\alpha = -\nabla L \cdot \nabla S$ is the progressive sharpening coefficient and $\beta = \|\nabla S\|^2$.

### 4.2 ANALYZING THE SIMPLIFIED DYNAMICS

We now analyze the dynamics in eq. (3). First, note that $x_t$ changes sign at every iteration, and that, $x_{t+1} \approx -x_t$ due to the instability in the $u$ direction. While eq. (3) cannot be directly modeled by an ODE due to these rapid oscillations, we can instead model $|x_t|, y_t$, whose update is controlled by $\eta$. As a consequence, we can couple the dynamics of $|x_t|, y_t$ to the following ODE $X(t), Y(t)$:

$$X'(t) = X(t) Y(t) \quad \text{and} \quad Y'(t) = \alpha - \beta \frac{X(t)^2}{2}. \tag{4}$$

This system has the unique fixed point $(X, Y) = (\delta, 0)$ where $\delta := \sqrt{2\alpha/\beta}$. We also note that this ODE can be written as a Lotka-Volterra predator-prey model after a change of variables, which is a classical example of a negative feedback loop. In particular, the following quantity is conserved:

**Lemma 3.** *Let* $h(z) := z - \log z - 1$. *Then* $g(X(t), Y(t)) := h\left( \frac{\beta X(t)^2}{2\alpha} \right) + \frac{Y(t)^2}{\alpha}$ *is conserved.*

*Proof.* $\frac{d}{dt} g(X(t), Y(t)) = \frac{\beta X(t)^2 Y(t)}{\alpha} - 2Y(t) + \frac{2}{\alpha} Y(t) \left[ \alpha - \beta \frac{X(t)^2}{2} \right] = 0.$ □

As a result we can use the conservation of $g$ to explicitly bound the size of the trajectory:

**Corollary 1.** *For all $t$, $X(0) \leq X(t) \lesssim \delta \sqrt{\log(\delta/X(0))}$ and $|Y(t)| \lesssim \sqrt{\alpha \log(\delta/X(0))}$.*

The fluctuations in sharpness are $\tilde{O}(\sqrt{\alpha})$, while the fluctuations in the unstable direction are $\tilde{O}(\delta)$. Moreover, the *normalized* displacement in the $\nabla S$ direction, i.e. $\frac{\nabla S}{\|\nabla S\|} \cdot (\theta - \theta^\star)$ is also bounded by $\tilde{O}(\delta)$, so the entire process remains bounded by $\tilde{O}(\delta)$. Note that the fluctuations increase as the progressive sharpening constant $\alpha$ grows, and decrease as the self-stabilization force $\beta$ grows.

## 4.3 Relationship with the constrained trajectory $\theta_t^\dagger$

Equation (3) completely determines the displacement $\theta_t - \theta^\star$ in the $u, \nabla S$ directions and Section 4.2 shows that these dynamics remain bounded by $\tilde{O}(\delta)$ where $\delta = \sqrt{2\alpha/\beta}$. However, progress is still made in all other directions. Indeed, $\theta_t$ evolves in these orthogonal directions by $-\eta P_{u,\nabla S}^\perp \nabla L$ at every step where $P_{u,\nabla S}^\perp$ is the projection onto this orthogonal subspace. This can be interpreted as first taking a gradient step of $-\eta \nabla L$ and then projecting out the $\nabla S$ direction to ensure the sharpness does not change. Lemma 13, given in the Appendix, shows that this is precisely the update for $\theta_t^\dagger$ (eq. (2)) up to higher order terms. The preceding derivation thus implies that $\|\theta_t - \theta_t^\dagger\| \leq \tilde{O}(\delta)$ and that this $\tilde{O}(\delta)$ error term is controlled by the self-stabilizing dynamics in eq. (3).

# 5 The Predicted Dynamics and Theoretical Results

We now present the equations governing edge of stability for general loss functions.

## 5.1 Notation

Our general approach Taylor expands the gradient of each iterate $\theta_t$ around the corresponding iterate $\theta_t^\dagger$ of the constrained trajectory. We define the following Taylor expansion quantities at $\theta_t^\dagger$:

**Definition 2** (Taylor Expansion Quantities at $\theta_t^\dagger$)**.**

$$\nabla L_t := \nabla L(\theta_t^\dagger), \quad \nabla^2 L_t := \nabla^2 L(\theta_t^\dagger), \quad \nabla^3 L_t := \nabla^3 L(\theta_t^\dagger), \quad \nabla S_t := \nabla S(\theta_t^\dagger), \quad u_t := u(\theta_t^\dagger).$$

*Furthermore, for any vector-valued function $v(\theta)$, we define $v_t^\perp := P_{u_t}^\perp v(\theta_t^\dagger)$ where $P_{u_t}^\perp$ is the projection onto the orthogonal complement of $u_t$.*

We also define the following quantities which govern the dynamics near $\theta_t^\star$.

**Definition 3.** *Let $\alpha_t := -\nabla L_t \cdot \nabla S_t$, $\beta_t := \left\|\nabla S_t^\perp\right\|^2$, and $\delta_t := \sqrt{\frac{2\alpha_t}{\beta_t}}$. Furthermore, we define*

$$\beta_{s\to t} := \nabla S_{t+1}^\perp \left[\prod_{k=t}^{s+1}(I - \eta\nabla^2 L_k)P_{u_k}^\perp\right]\nabla S_s^\perp \text{ and } \delta := \sup_t \delta_t.$$

Recall that $\alpha_t$ is the progressive sharpening force, $\beta_t$ is the strength of the stabilization force, and $\delta_t$ controls the size of the deviations from $\theta_t^\dagger$ and was the fixed point in the $x$ direction in Section 4.2. The scalars $\beta_{s\to t}$ capture the effect of the interactions between $\nabla S$ and the Hessian.

## 5.2 The equations governing edge of stability

We now introduce the equations governing edge of stability. We track the following quantities:

**Definition 4.** *Define $v_t := \theta_t - \theta_t^\dagger$, $x_t := u_t \cdot v_t$, $y_t := \nabla S_t^\perp \cdot v_t$.*

Our predicted dynamics directly predict the displacement $v_t$ and the full definition is deferred to Appendix C. However, they have a relatively simple form in the $u_t, \nabla S_t^\perp$ directions:

**Lemma 4** (Predicted Dynamics for $x, y$)**.** *Let $\mathring{v}_t$ denote our predicted dynamics (defined in Appendix C). Letting $\mathring{x}_t = u_t \cdot \mathring{v}_t$ and $\mathring{y}_t = \nabla S_t^\perp \cdot \mathring{v}_t$, we have*

$$\mathring{x}_{t+1} = -(1 + \eta\mathring{y}_t)\mathring{x}_t \quad and \quad \mathring{y}_{t+1} = \eta\sum_{s=0}^{t}\beta_{s\to t}\left[\frac{\delta_s^2 - \mathring{x}_s^2}{2}\right]. \tag{5}$$

Note that when $\beta_{s\to t}$ are constant, our update reduces to the simple case discussed in Section 4, which we analyze fully. When $x_t$ is large, eq. (5) demonstrates that there is a self-stabilization force which acts to decrease $y_t$; however, unlike in Section 4, the strength of this force changes with $t$.

### 5.3 COUPLING THEOREM

We now show that, under a mild set of assumptions which we verify to hold empirically in Appendix E, the true dynamics are accurately governed by the predicted dynamics. This lets us use the predicted dynamics to predict the loss, sharpness, and the distance to the constrained trajectory $\theta_t^\dagger$.

Our errors depend on the unitless quantity $\epsilon$, which we verify is small in Appendix E.

**Definition 5.** *Let $\epsilon_t := \eta\sqrt{\alpha_t}$ and $\epsilon := \sup_t \epsilon_t$.*

To control Taylor expansion errors, we require upper bounds on $\nabla^3 L$ and its Lipschitz constant:[5]

**Assumption 3.** *Let $\rho_3$, $\rho_4$ to be the minimum constants such that for all $\theta$, $\left\|\nabla^3 L(\theta)\right\|_{op} \leq \rho_3$ and $\nabla^3 L$ is $\rho_4$-Lipschitz with respect to $\|\cdot\|_{op}$. Then we assume that $\rho_4 = O(\eta\rho_3^2)$.*

Next, we require the following generalization of Assumption 1:

**Assumption 4.** *For all $t$, $\frac{-\nabla L_t \cdot \nabla S_t}{\|\nabla L_t\|\|\nabla S_t^\perp\|} = \Theta(1)$ and $\left\|\nabla S_t^\perp\right\| = \Theta(\rho_3)$.*

Finally, we require a set of "non-worst-case" assumptions, which are that the quantities $\nabla^2 L$, $\nabla^3 L$, and $\lambda_{min}(\nabla^2 L)$ are nicely behaved in the directions orthogonal to $u_t$, which generalizes the eigengap assumption. We verify the assumptions on $\nabla^2 L$ and $\nabla^3 L$ empirically in Appendix E.

**Assumption 5.** *For all $t$ and $v, w \perp u_t$, $\frac{\left\|\nabla^3 L_t(v,w)\right\|}{\|\nabla^3 L_t\|_{op}\|v\|\|w\|}$, $\frac{|\nabla^2 L_t(\mathring{v}_t^\perp, \mathring{v}_t^\perp)|}{\|\nabla^2 L_t\|\|\mathring{v}_t^\perp\|^2}$, $\frac{|\lambda_{min}(\nabla^2 L_t)|}{\|\nabla^2 L_t\|_2} \leq O(\epsilon)$.*

With these assumptions in place, we can state our main theorem which guarantees $\mathring{x}, \mathring{y}, \mathring{v}$ predict the loss, sharpness, and deviation from the constrained trajectory up to higher order terms:

**Theorem 1.** *Let $\mathscr{T} := O(\epsilon^{-1})$ and assume that $\min_{t \leq \mathscr{T}} |\mathring{x}_t| \geq c_1\delta$. Then for any $t \leq \mathscr{T}$, we have*

$$L(\theta_t) = L(\theta_t^\dagger) + \mathring{x}_t^2/\eta + O\left(\epsilon\delta^2/\eta\right) \tag{Loss}$$

$$S(\theta_t) = 2/\eta + \mathring{y}_t + (S_t \cdot u_t)\mathring{x}_t + O\left(\epsilon^2/\eta\right) \tag{Sharpness}$$

$$\theta_t = \theta_t^\dagger + \mathring{v}_t + O(\epsilon\delta) \tag{Deviation from $\theta^\dagger$}$$

The sharpness is controlled by the slowly evolving quantity $\mathring{y}_t$ and the period-2 oscillations of $(\nabla S \cdot U)\mathring{x}_t$. This combination of gradual and rapid periodic behavior was observed by Cohen et al. (2021) and appears in our experiments. Theorem 1 also shows that the loss at $\theta_t$ spikes whenever $\mathring{x}_t$ is large. On the other hand, when $\mathring{x}_t$ is small, $L(\theta_t)$ approaches the loss of the constrained trajectory.

## 6 EXPERIMENTS

We verify that the predicted dynamics defined in eq. (5) accurately capture the dynamics of gradient descent at the edge of stability by replicating the experiments in (Cohen et al., 2021) and tracking the deviation of gradient descent from the constrained trajectory. In Figure 3, we evaluate our theory on a 3-layer MLP and a 3-layer CNN trained with mean squared error (MSE) on a 5k subset of CIFAR10 and a 2-layer Transformer (Vaswani et al., 2017) trained with MSE on SST2 Socher et al. (2013). We provide additional experiments varying the learning rate and loss function in Appendix G, which use the generalized predicted dynamics described in Section 7.2. For additional details, see Appendix D.

Figure 3 confirms that the predicted dynamics eq. (5) accurately predict the loss, sharpness, and distance from the constrained trajectory. In addition, while the gradient flow trajectory diverges from the gradient descent trajectory at a linear rate, the gradient descent trajectory and the constrained trajectories remain close *throughout training*. In particular, the dynamics converge to the fixed point $(|x_t|, y_t) = (\delta_t, 0)$ described in Section 4.2 and $\|\theta_t - \theta_t^\dagger\| \to \delta_t$. This confirms our claim that gradient descent implicitly follows the constrained trajectory eq. (2).

In Section 5, various assumptions on the model were made to obtain the EOS behavior. In Appendix E, we numerically verify these assumptions to ensure the validity of our theory.

---

[5]For simplicity of exposition, we make these bounds on $\nabla^3 L$ globally, however our proof only requires them in a small neighborhood of the constrained trajectory $\theta^\dagger$.

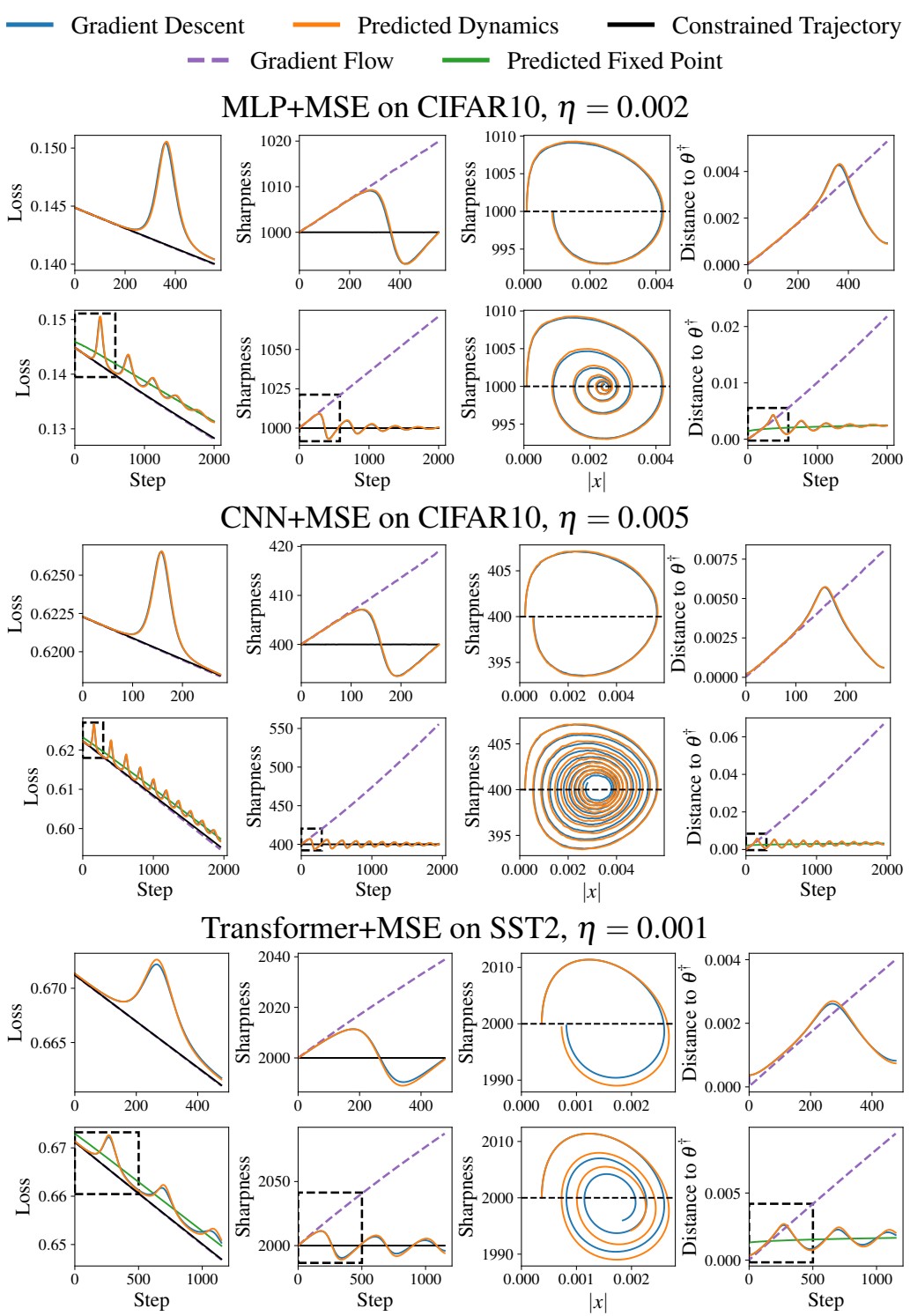

Figure 3: We empirically demonstrate that the predicted dynamics given by eq. (5) track the true EOS dynamics. For each learning rate, the top row is a zoomed in copy of the bottom row which isolates one cycle and is represented by the dashed rectangle. Reported sharpnesses are two-step averages for visual clarity. For additional experimental details, see Section 6 and Appendix D.

## 7 DISCUSSION

### 7.1 TAKEAWAYS FROM THE PREDICTED DYNAMICS

A key consequence of the predicted dynamics is that the loss and sharpness only depend on $(\mathring{x}_t, \mathring{y}_t)$, which are governed by the 2D dynamical system eq. (5). Therefore, understanding the EOS dynamics only requires analyzing this dynamical system, which is generally well behaved (Figure 3). Furthermore, we expect $\mathring{x}_t, \mathring{y}_t$ to converge to $(\pm\delta_t, 0)$, the fixed point of eq. (5). In fact, Figure 3 shows that after a few cycles, $(\mathring{x}_t, \mathring{y}_t)$ indeed converges to this fixed point. We can accurately predict its location as well as the loss increase from the constrained trajectory due to $\mathring{x}_t \neq 0$.

### 7.2 GENERALIZED PREDICTED DYNAMICS

In order for our cubic Taylor expansions to track the true gradients, we require a bound on the fourth derivative of the loss (Assumption 3). This is usually sufficient to capture the EOS dynamics as demonstrated by Figure 3 and Appendix E. However, this condition was violated in some of our experiments, especially when using logistic loss. To overcome this challenge, we developed a generalized form of the predicted dynamics whose definition we defer to Appendix F. These generalized predictions are qualitatively similar to those given by the predicted dynamics in Section 5; however, they precisely track the dynamics of gradient descent in a wider range of settings (see Appendix G).

### 7.3 IMPLICATIONS FOR NEURAL NETWORK TRAINING

**Non-Monotonic Loss Decrease**  An important property of EOS is that the loss decreases over long time scales, albeit non-monotonically. Our theory provides a clear explanation for this phenomenon. We show that the gradient descent trajectory remains close to the constrained trajectory (Sections 4 and 5). Since the constrained trajectory is *stable*, it satisfies a descent lemma (Lemma 14), and its loss monotonically decreases. Over short time periods, the loss is dominated by the rapid fluctuations of $x_t$ described in Section 4. Over longer time periods, the loss decrease of the constrained trajectory overpowers the bounded fluctuations of $x_t$, leading to an overall loss decrease.

**Generalization & the Role of Large Learning Rates**  Prior work has shown that in neural networks, both decreasing sharpness of the learned solution (Keskar et al., 2017; Dziugaite & Roy, 2017; Neyshabur et al., 2017; Jiang et al., 2020) and increasing the learning rate (Smith et al., 2018; Li et al., 2019; Lewkowycz et al., 2020) are correlated with better generalization. Our analysis shows that gradient descent implicitly constrains the sharpness to stay below $2/\eta$, which suggests larger learning may improve generalization by reducing the sharpness. In Figure 4 we confirm that in a standard setting, full-batch gradient descent generalizes better with large learning rates.

**Training Speed**  Additional experiments in (Cohen et al., 2021, Appendix F) show that, despite the instability in the training process, larger learning rates lead to faster convergence. This phenomenon is explained by our analysis. Gradient descent is coupled to the constrained trajectory which minimizes the loss while constraining movement in the $u_t, \nabla S_t^\perp$ directions. Since only two directions are "off limits," the constrained trajectory can still move quickly in the orthogonal directions, using the large learning rate to accelerate convergence. We demonstrate this empirically in Figure 4.

We defer additional discussion of our work, including the effect of multiple unstable eigenvalues and connections to Sharpness Aware Minimization (Foret et al., 2021), warm-up (Gilmer et al., 2021), and scale-invariant loss functions (Lyu et al., 2022) to Appendix H.

### 7.4 FUTURE WORK

An important direction for future work is understanding the EOS dynamics when there are multiple unstable eigenvalues, which we briefly discuss in Appendix H. Another interesting direction is understanding the *global* convergence properties at EOS, including convergence to a KKT point of the constrained update eq. (2). Next, our analysis focused on the EOS dynamics but left open the question of why neural networks exhibit progressive sharpening. Finally, we would like to understand the role of self-stabilization in *stochastic*-gradient descent and how it interacts with the implicit biases of SGD (Blanc et al., 2020; Damian et al., 2021; Li et al., 2022a).

ACKNOWLEDGEMENTS

AD acknowledges support from a NSF Graduate Research Fellowship. EN acknowledges support from a National Defense Science & Engineering Graduate Fellowship, and NSF grants CIF-1907661 and DMS-2014279. JDL, AD, EN acknowledge support of the Sloan Research Fellowship, NSF CCF 2002272, NSF IIS 2107304, NSF CIF 2212262, and NSF-CAREER under award #2144994.

The authors would like to thank Jeremy Cohen, Kaifeng Lyu, and Lei Chen for helpful discussions throughout the course of this project. We would especially like to thank Jeremy Cohen for suggesting the term "self-stabilization" to describe the negative feedback loop derived in this paper.

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

TABLE OF CONTENTS

## A   NOTATION

We denote by $\nabla^k L(\theta)$ the $k$-th order derivative of the loss $L$ at $\theta$. Note that $\nabla^k L(\theta)$ is a symmetric $k$-tensor in $(\mathbb{R}^d)^{\otimes k}$ when $\theta \in \mathbb{R}^d$.

For a symmetric $k$-tensor $T$, and vectors $u_1, \ldots, u_j \in \mathbb{R}^d$ we will use $T(u_1, \ldots, u_j)$ to denote the tensor contraction of $T$ with $u_1, \ldots, u_j$, i.e.

$$[T(u_1, \ldots, u_k)]_{i_1, \ldots, i_{k-j}} := T_{i_1, \ldots, i_k} (u_1)_{i_{k-j+1}} \cdots (u_j)_{i_k}.$$

We use $P_{u_1, \ldots, u_k}$ to denote the orthogonal projection onto $\mathrm{span}(u_1, \ldots, u_k)$ and $P^\perp_{u_1, \ldots, u_k}$ is the projection onto the corresponding orthogonal complement.

For matrices $A_1, \ldots, A_k$, we define

$$\prod_{k=1}^{t} A_k := A_1 \ldots A_t \quad \text{and} \quad \prod_{k=t}^{1} A_k := A_t \ldots A_1.$$

## B   A TOY MODEL FOR SELF-STABILIZATION

For $\alpha, \beta > 0$, consider the function

$$L(x, y, z) := \left( \frac{2}{\eta} + \sqrt{\beta} y \right) \frac{x^2}{2} - \frac{\alpha}{\sqrt{\beta}} y - z$$

initialized at the point $(x_0, 0, 0)$. Note that the constrained trajectory will follow $x_t^\dagger = 0$, $y_t^\dagger = 0$, $z_t^\dagger = -\eta t$ as it cannot decrease $y$ without increasing the sharpness past $2/\eta$. We therefore have:

$$\nabla L_t = \left[ 0, -\frac{\alpha}{\sqrt{\beta}}, 1 \right], \ u_t = [1, 0, 0], \ S_t = 2/\eta + \sqrt{\beta} y, \ \nabla^2 L_t = S_t u_t u_t^t, \ \nabla S_t = \left[ 0, \sqrt{\beta}, 0 \right].$$

Note that this satisfies all of the assumptions in Section 4 and it satisfies $\alpha = -\nabla L_t \cdot \nabla S_t$ and $\beta = \|\nabla S_t\|^2$. This process will then follow eq. (4) in the $x, y$ directions while it tracks the constrained trajectory $\theta_t^\dagger$ moving linearly in the $-P^\perp_{u, \nabla S} \nabla L = [0, 0, -1]$ direction.

## C   DEFINITION OF THE PREDICTED DYNAMICS

Below, we present the full definition of the predicted dynamics:

**Definition 6** (Predicted Dynamics, full)**.** *Define $\mathring{v}_0 = v_0$, and let $\mathring{x}_t = \mathring{v}_t \cdot u_t, \mathring{y}_t = \nabla S^\perp \cdot \mathring{v}_t$. Then*

$$v_{t+1}^* = P_{u_{t+1}}^\perp (I - \eta \nabla^2 L_t) P_{u_t}^\perp v_t^* + \eta P_{u_{t+1}}^\perp \nabla S_t^\perp \left[ \frac{\delta_t^2 - x_t^{*2}}{2} \right] - (1 + \eta y_t^*) x_t^* \cdot u_{t+1} \tag{6}$$

For convenience, we will define the map $\text{step}_t : \mathbb{R}^d \to \mathbb{R}^d$ as follows:

**Definition 7.** *Given a vector $v$ and a timestep $t$, define $\text{step}_t(v)$ by*

$$P_{u_{t+1}}^\perp \text{step}_t(v) = P_{u_{t+1}}^\perp \left[ (I - \eta \nabla^2 L_t) P_{u_t}^\perp v + \eta \nabla S_t^\perp \left[ \frac{\delta_t^2 - x^2}{2} \right] \right] \tag{7}$$

$$u_{t+1} \cdot \text{step}_t(v) = -(1 + \eta y) x. \tag{8}$$

*where $x = u_t \cdot v$ and $y = \nabla S_t^\perp \cdot v$.*

It is easy to see that $\mathring{v}_{t+1} = \text{step}_t(\mathring{v}_t)$.

*Proof of Lemma 4.* Defining $A_t = (I - \eta \nabla^2 L_t) P_{u_t}^\perp$, we can unfold the recursion in *eq.* (6) to obtain the following formula for $\mathring{v}_t$.

$$v_{t+1}^* = \eta \sum_{s=0}^t P_{u_{t+1}}^\perp \left[ \prod_{k=t}^{s+1} A_k \right] \nabla S_s^\perp \left[ \frac{\delta_s^2 - x_s^{*2}}{2} \right] - (1 + \eta y_t^*) x_t^* \cdot u_{t+1}. \tag{9}$$

It is then immediate to see that $\mathring{x}_t = \mathring{v}_t \cdot u_t, \mathring{y}_t = \nabla S_t^\perp \cdot \mathring{v}_t$ have the following simple update:

$$x_{t+1}^* = -(1 + \eta y_t^*) x_t^* \quad \text{and} \quad y_{t+1}^* = \eta \sum_{s=0}^t \beta_{s \to t} \left[ \frac{\delta_s^2 - x_s^{*2}}{2} \right],$$

where we recall that we have defined

$$\beta_{s \to t} := \nabla S_{t+1}^\perp \left[ \prod_{k=t}^{s+1} A_k \right] \nabla S_s^\perp. \tag{10}$$

$\square$

# D    EXPERIMENTAL DETAILS

## D.1    ARCHITECTURES

We evaluated our theory on four different architectures. The 3-layer MLP and CNN are exact copies of the MLP and CNN used in (Cohen et al., 2021). The MLP has width 200, the CNN has width 32, and both are using the swish activation (Ramachandran et al., 2017). We also evaluate on a ResNet18 with progressive widths 16, 32, 64, 128 and on a 2-layer Transformer with hidden dimension 64 and two attention heads.

## D.2    DATA

We evaluated our theory on three primary tasks: CIFAR10 multi-class classification with both categorical MSE loss and cross-entropy loss, CIFAR10 binary classification (cats vs dogs) with binary MSE loss and logistic loss, and SST2 (Socher et al., 2013) with binary MSE loss and logistic loss.

## D.3    EXPERIMENTAL SETUP

For every experiment, we tracked the gradient descent dynamics until they reached instability and then began tracking the constrained trajectory, gradient descent, gradient flow, and both our predicted dynamics (Section 5) and our generalized predicted dynamics (Appendix F). In addition, we tracked the various quantities on which we made assumptions for Section 5 in order to validate these assumptions. We also tracked the second eigenvalue of the Hessian at the constrained trajectory

throughout training and stopped training once it reached $1.9/\eta$, to ensure the existence of a single unstable eigenvalue. Finally, as the edge of stability dynamics are very sensitive to small perturbation when $|x|$ is small (see Figure 2), we switched to computing gradients with 64-bit precision after first reaching instability to avoid propagating floating point errors.

Eigenvalues were computed using the LOBPCG sparse eigenvalue solver in JAX (Bradbury et al., 2018). To compute the constrained trajectory, we computed a linearized approximation for $\mathrm{proj}_{\mathcal{M}}$ inspired by Lemma 13 along with a Newton step in the $u_t$ direction to ensure that $\nabla L \cdot u = 0$. Each linearized approximation step required recomputing the sharpness and top eigenvector and each projection step then consisted of three linearized projection steps, for a total of three eigenvalue computations per projection step.

Our experiments were conducted in JAX (Bradbury et al., 2018), using `https://github.com/locuslab/edge-of-stability` as a reference for replicating the experimental setup used in (Cohen et al., 2021). All experiments were conducted on two servers, each with 10 NVIDIA GPUs. Our code can be found at `https://github.com/adamian98/EOS`.

## E    EMPIRICAL VERIFICATION OF THE ASSUMPTIONS

For each of the experimental settings considered (MLP+MSE, CNN+MSE, CNN+Logistic, ResNet18+MSE, Transformer+MSE, Transformer+Logistic), we plot a number of quantities along the constrained trajectory to verify that the assumptions made in the main text hold. For each learning rate $\eta$ we have 8 plots tracking various quantities, which verify the assumptions as follows: Assumption 1 is verified by the 1st plot, $\epsilon$ being small is verified by the 2nd plot, Assumption 4 is verified by the 3rd and 4th plots, Assumption 3 is verified by the 5th plot, and Assumption 5 is verified by the last 3 plots. As described in the experimental setup, training is stopped once the second eigenvalue is $1.9/\eta$, so Assumption 2 always holds with $c = 1.9$ as well.

# MLP+MSE on CIFAR10

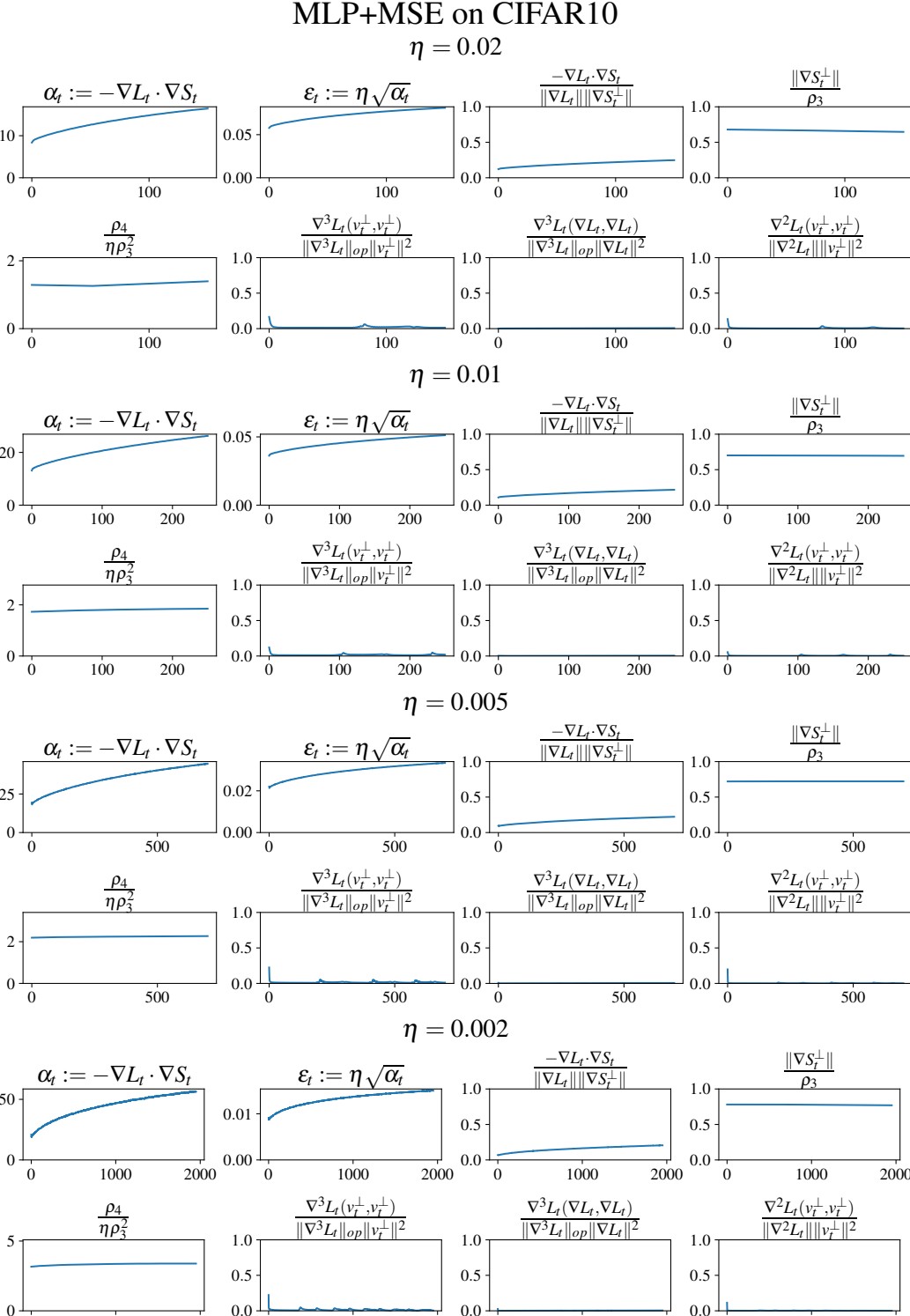

# CNN+MSE on CIFAR10

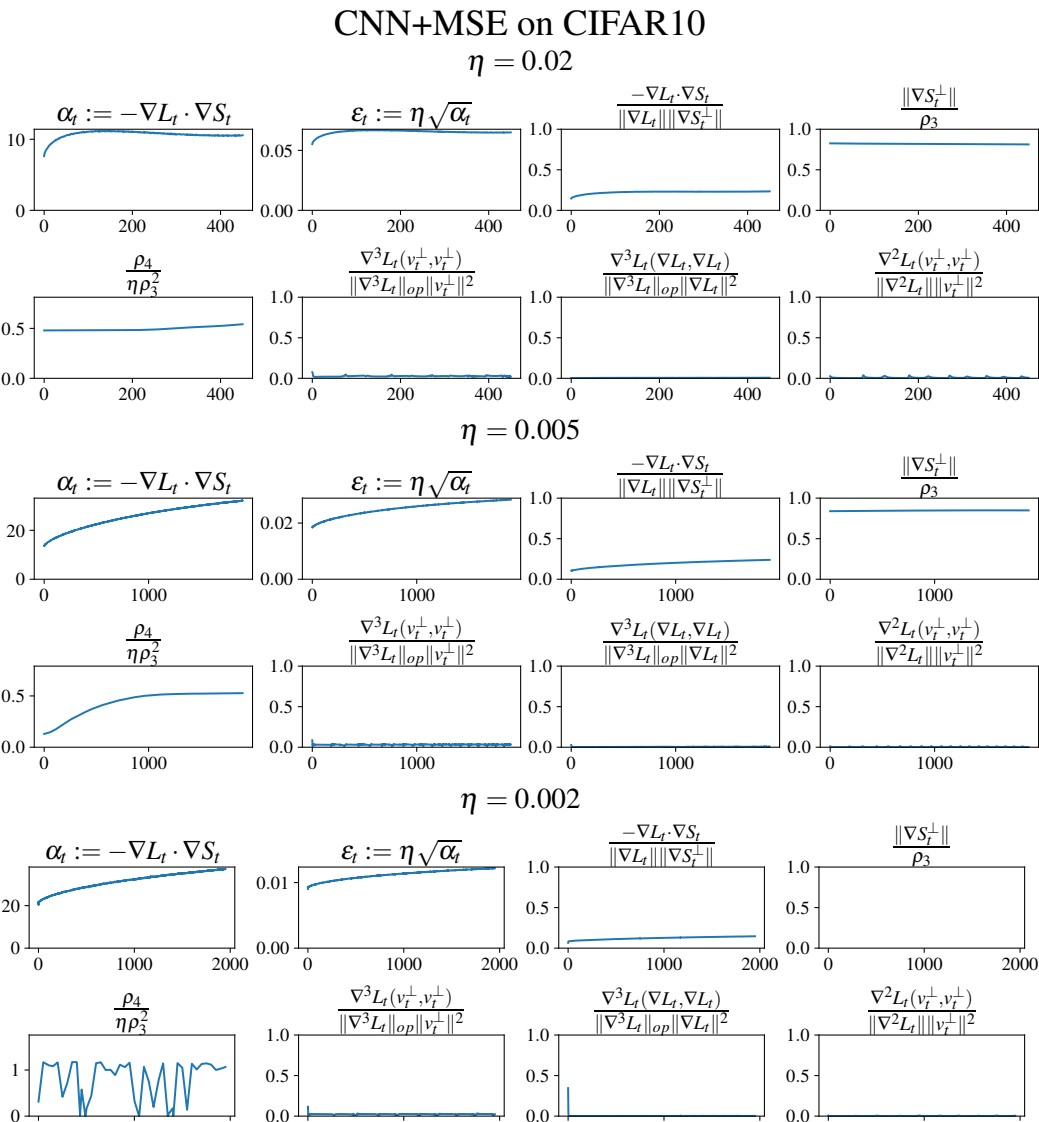

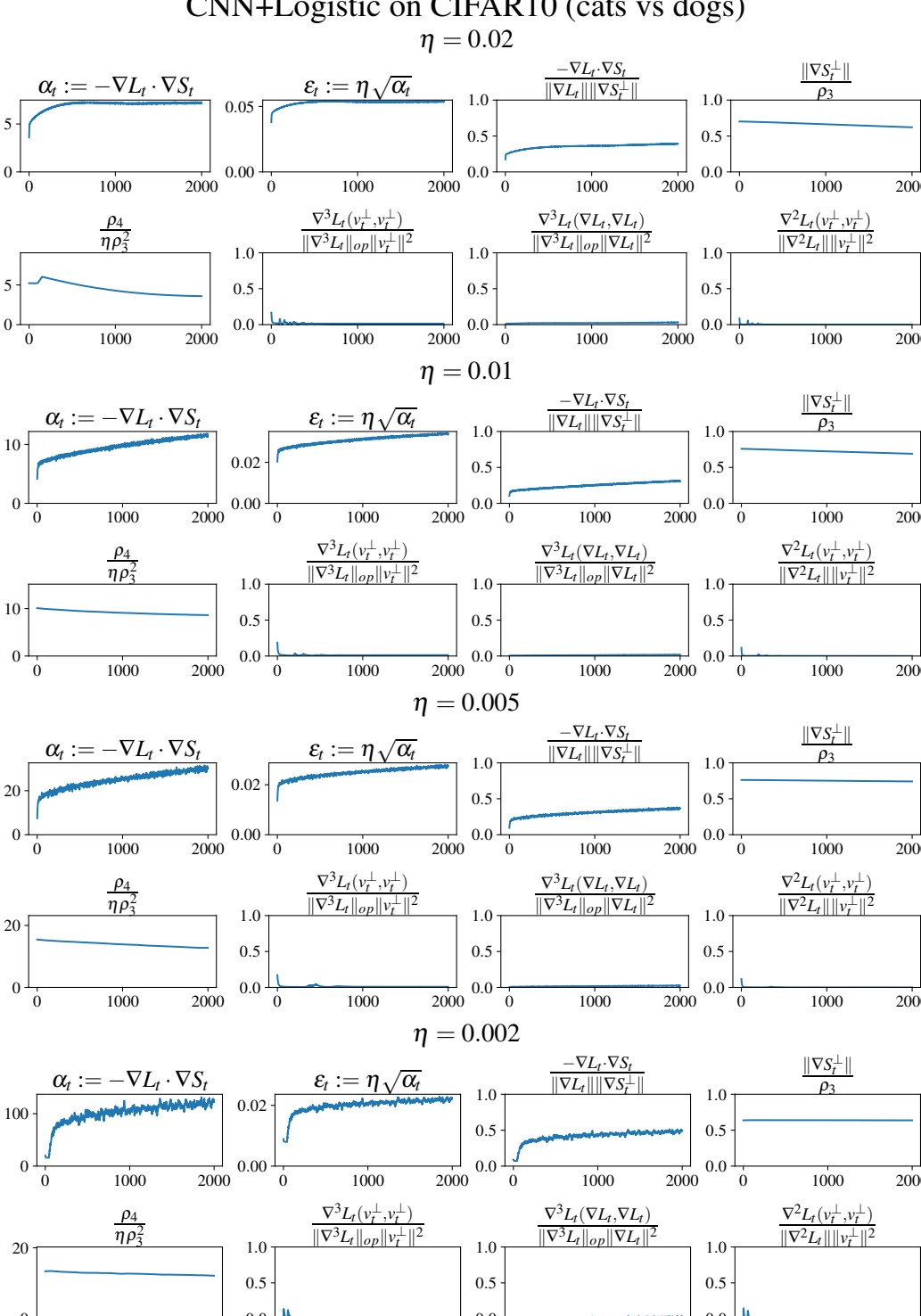

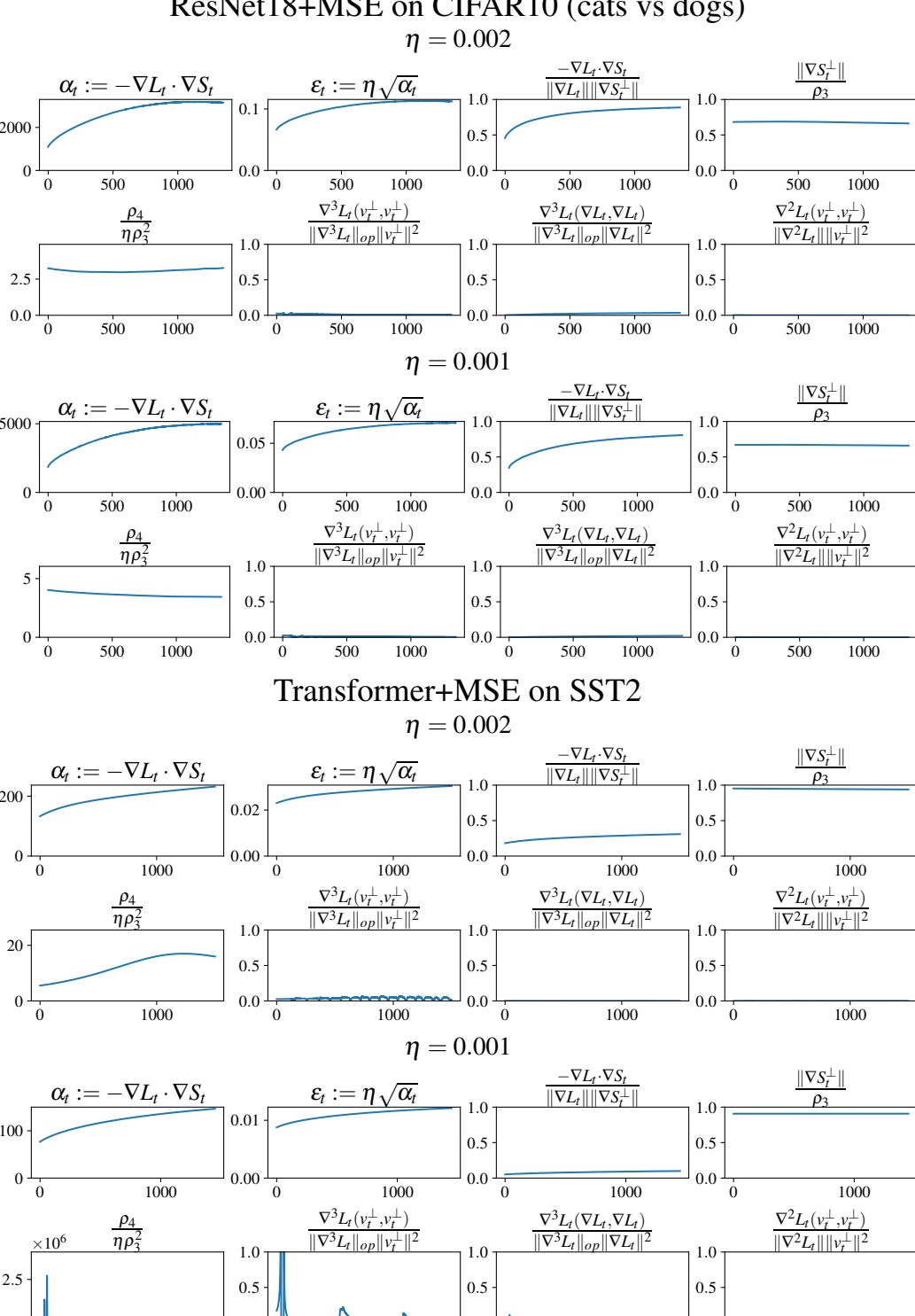

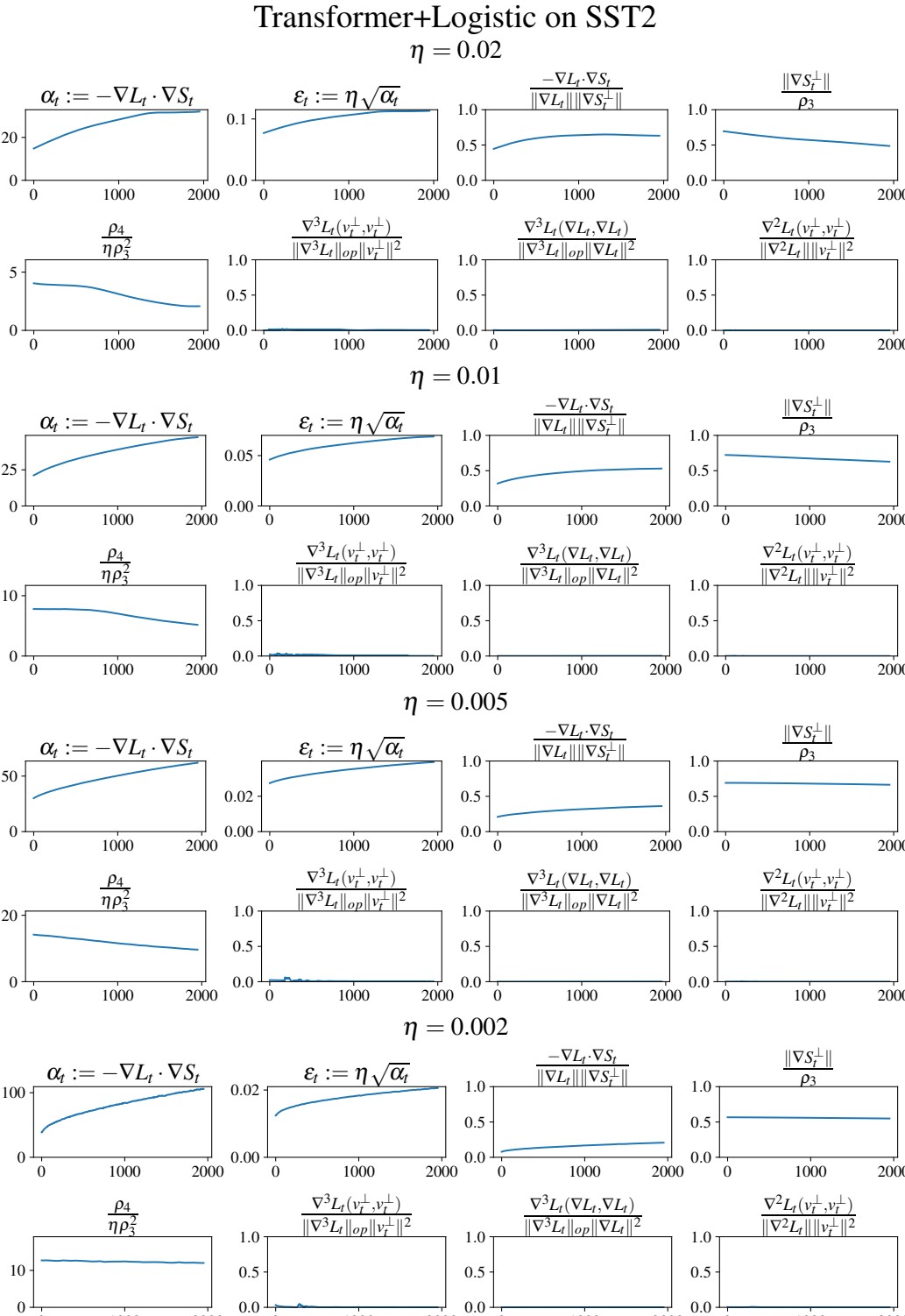

## F   The Generalized Predicted Dynamics

Our analysis relies on a cubic Taylor expansion of the gradient. However, in order for this Taylor expansion to accurately track the gradients we need a bound on the fourth derivative of the loss (Assumption 3). Section 6 and Appendix E show that this approximation is sufficient to capture the dynamics of gradient descent at the edge of stability for many standard models when the loss criterion is the mean squared error. However, for certain architectures and loss functions, including ResNet18 and models trained with the logistic loss, this condition is often violated.

In these situations, the loss function in the top eigenvector direction is either *sub-quadratic*, meaning that the quadratic Taylor expansion overestimates the loss and sharpness[6], or *super-quadratic*, meaning that the quadratic Taylor expansion underestimates the loss and sharpness. To capture this phenomenon, we derive a more general form of the predicted dynamics which reduces to the standard predicted dynamics in Section 5 when the loss in the top eigenvector direction is approximately quadratic. In addition, Appendix G shows that the generalized predicted dynamics capture the dynamics of gradient descent at the edge of stability for both mean squared error and cross-entropy in all settings we tested.

### F.1   Deriving the Generalized Predicted Dynamics

To derive the generalized predicted dynamics, we will abstract away the dynamics in the top eigenvector direction. Specifically, for every $t$ we define

$$F_t(x) := L(\theta_t^\dagger + x u_t) - L(\theta_t^\dagger) - \frac{x^2}{\eta}.$$

We say that $L$ is sub-quadratic at $t$ if $F_t(x) < 0$ and super-quadratic if $F_t(x) > 0$.

Note that knowing $F_t$ is not sufficient to capture the dynamics in the $u_t$ direction. Specifically,

$$x_{t+1} = x_t - \eta u_t \cdot \nabla L(\theta_t^\dagger + v_t) \neq x_t - \eta u_t \cdot \nabla L(\theta_t^\dagger + x u_t).$$

It is still critically important to track the effect that the movement in the $\nabla S_t^\perp$ direction has on the dynamics of $x$. As in Section 4.1, the effect of the movement in the $\nabla S_t^\perp$ direction on the dynamics of $x$ is changing the sharpness by $y_t$. This gives us the generalized predicted dynamics update:

$$v_{t+1}^* = P_{u_{t+1}}^\perp (I - \eta \nabla^2 L_t) P_{u_t}^\perp v_t^* + \eta P_{u_{t+1}}^\perp \nabla S_t^\perp \left[ \frac{\delta_t^2 - x_t^{*2}}{2} \right] - x_{t+1}^\star \cdot u_{t+1}$$

where $x_{t+1}^\star = -(1 + \eta y_t^\star) x_t^\star - \eta F'(x_t^\star)$.

Note that when $F_t(x) = 0$ is exactly quadratic, this reduces to the standard predicted dynamics update in eq. (6). Note that the update for $y$ is completely unchanged:

**Lemma 5.** *Restricted to the $u_t, \nabla S_t$ directions, the generalized predicted dynamics $v_t^\star$ imply:*

$$x_{t+1}^\star = -(1 + \eta y_t^\star) x_t^\star - \eta F'(x_t^\star) \quad and \quad y_{t+1}^\star = \eta \sum_{s=0}^{t} \beta_{s \to t} \left[ \frac{\delta_s^2 - x_s^{*2}}{2} \right]. \tag{11}$$

The proof is identical to the proof of Lemma 4.

### F.2   Properties of the Generalized Predicted Dynamics

Note that due to the sign flipping argument in Appendix I, we can assume that $F$ is an even function as the odd part will only influence the dynamics through additional oscillations of period 2, so throughout the remainder of this section we will assume that $F_t(x) = F_t(-x)$. Otherwise, we can simply redefine $F$ by its even part.

Next, note that the fixed point of eq. (11) is still when $x_t = \delta_t$, regardless of the shape of $F_t$, due to the need to stabilize the $\nabla S_t^\perp$ direction. This contradicts previous 1-dimensional analyses of edge

---

[6]This sub-quadratic phenomenon was also observed in (Ma et al., 2022).

of stability in which the fixed point in the top eigenvector direction strongly depends on the shape of $F_t$, the loss in the $u_t$ direction.

The limiting value of $y_t$ can therefore be read from the update for $x_t$. If $(\delta_t, y)$ is an orbit of period 2 of eq. (11), then

$$-\delta_t = -(1 + \eta y)\delta_t - \eta F'(\delta_t) \implies y = -\frac{F'(\delta_t)}{\delta_t}.$$

In addition, note that the sharpness can no longer be approximated as $S(\theta_t) \approx 2/\eta + y_t$ as the sharpness now changes along the $u_t$ direction. In particular, it changes by $F''(x)$ so that

$$S(\theta_t) \approx 2/\eta + y_t + F''(x_t).$$

Therefore, the limiting sharpness of eq. (11) is

$$S(\theta_t) \to 2/\eta - \frac{F'_t(\delta_t)}{\delta_t} + F''_t(\delta_t).$$

When $F_t = 0$ and the loss is exactly quadratic in the $u$ direction, this update reduces to fixed point predictions in Section 4.1.

One interesting phenomenon observed by Cohen et al. (2021) is that with cross-entropy loss, the sharpness was never exactly $2/\eta$, but usually hovered above it. This contradicts the predictions of the standard predicted dynamics which predict that the fixed point has sharpness 0. However, using the generalized predicted dynamics eq. (11), we can give a clear explanation.

When the loss is sub-quadratic, e.g. when $F_t(x) = -\rho_4 \frac{x^4}{24}$, we have

$$S(\theta_t) \to 2/\eta + \rho_4 \frac{\delta_t^2}{6} - \rho_4 \frac{\delta_t^2}{2} = 2/\eta - \rho_4 \frac{\delta_t^2}{3} < 2/\eta$$

so the sharpness will converge to a value *below* $2/\eta$. On the other hand if the loss is super-quadratic, the sharpness converges to a value *above* $2/\eta$. More generally, whether the loss converges to a value above or below $2/\eta$ depends on the sign of $F''_t(\delta_t) - \delta_t F'_t(\delta_t)$.

In our experiments in Appendix G, we observed both sub-quadratic and super-quadratic loss functions. In particular, the loss was usually sub-quadratic when it first reached instability but gradually became super-quadratic as training progressed at the edge of stability.

## G  ADDITIONAL EXPERIMENTS

### G.1  THE BENEFIT OF LARGE LEARNING RATES: TRAINING TIME AND GENERALIZATION

We trained ResNet18 with full batch gradient descent on the full 50k training set of CIFAR10 with various learning rates, in addition to the commonly proposed learning rate schedule $\eta_t := 1/S(\theta_t)$. We show that despite entering the edge of stability, large learning rates converge much faster. In addition, due to the self-stabilization effect of gradient descent, the final sharpness is bounded by $2/\eta$ which is smaller for larger learning rates and leads to better generalization (see Figure 4).

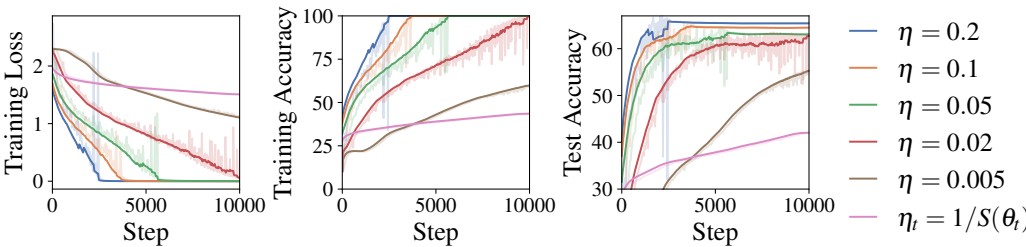

Figure 4: Large learning rates converge faster and generalize better (ResNet18 and CIFAR10).

## G.2 EXPERIMENTS WITH THE GENERALIZED PREDICTED DYNAMICS

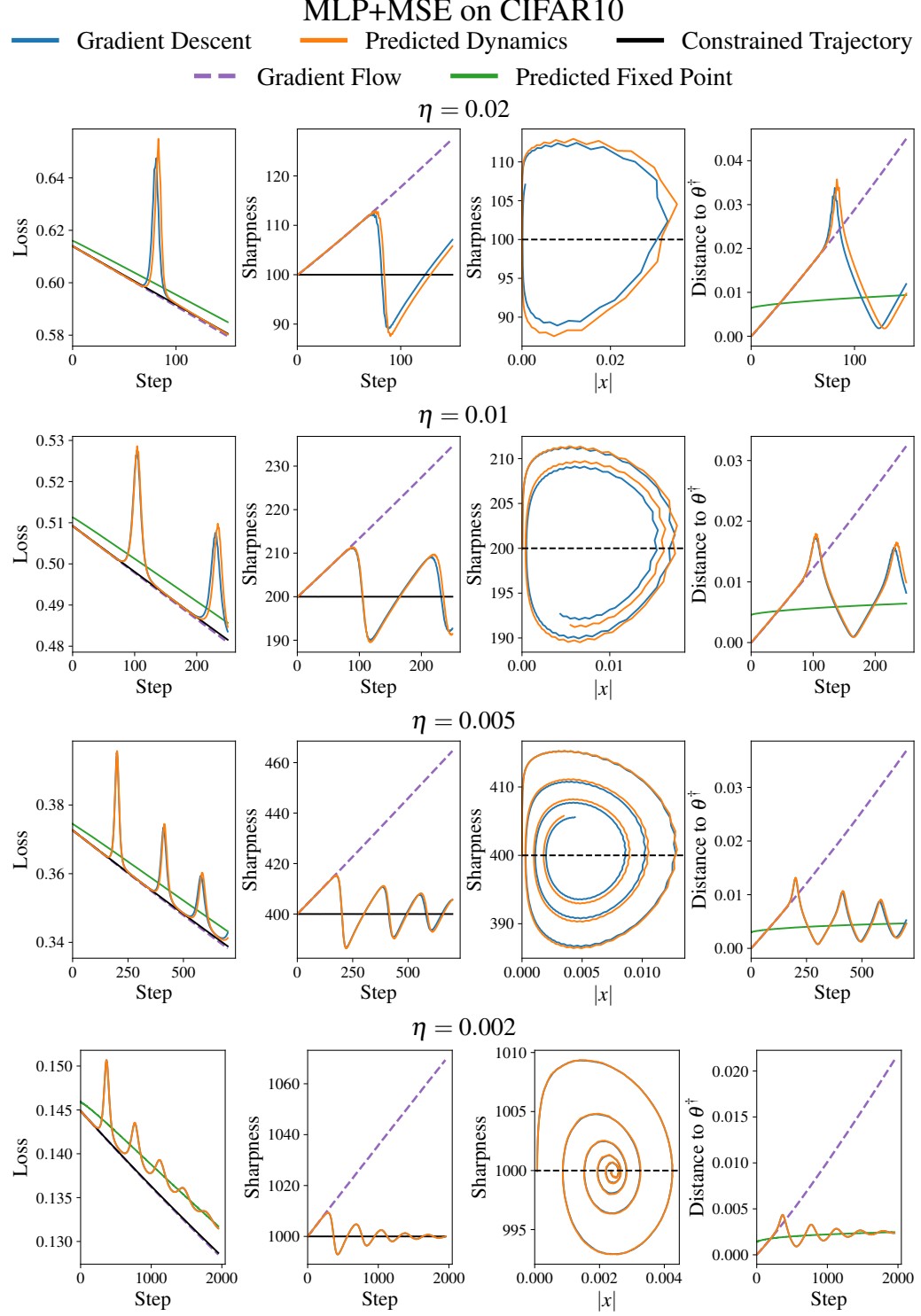

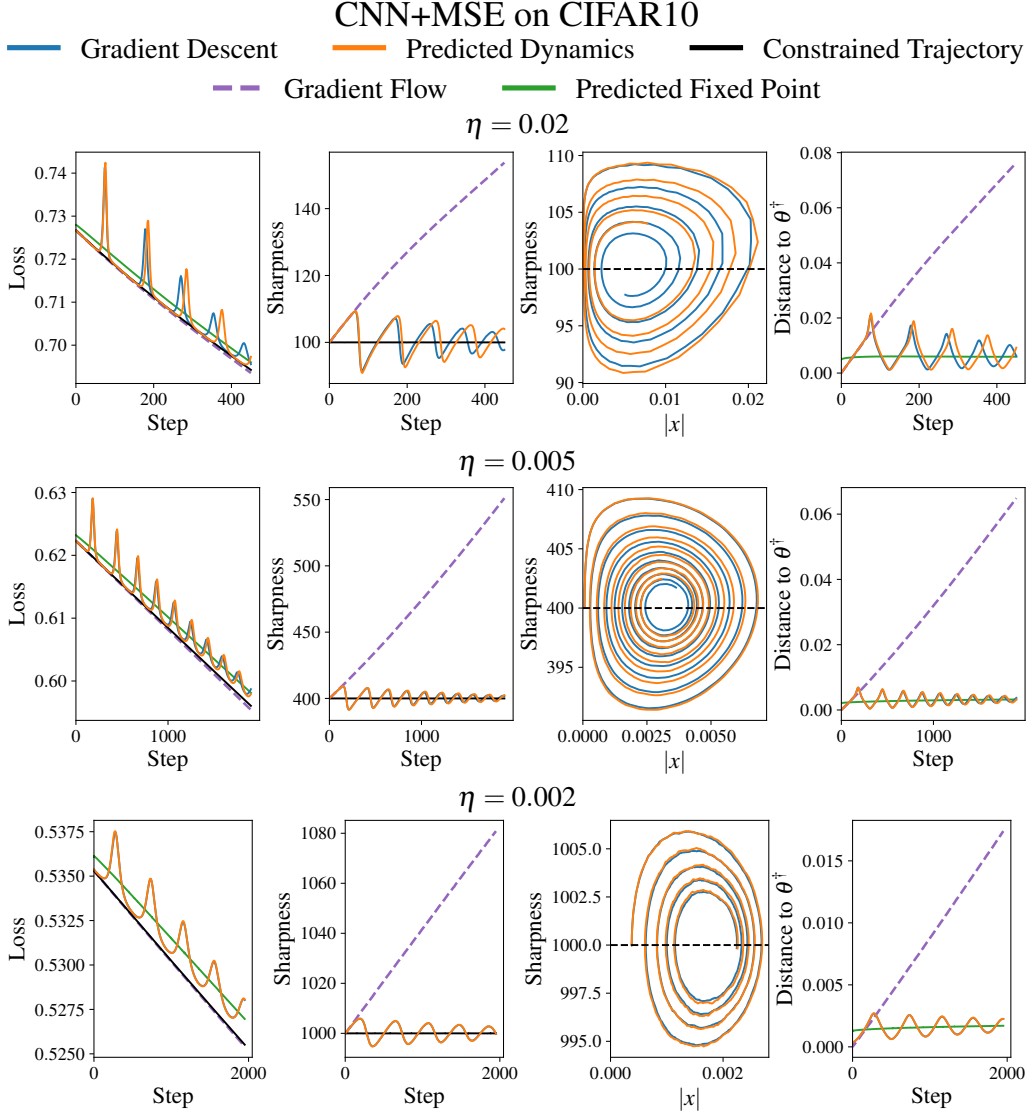

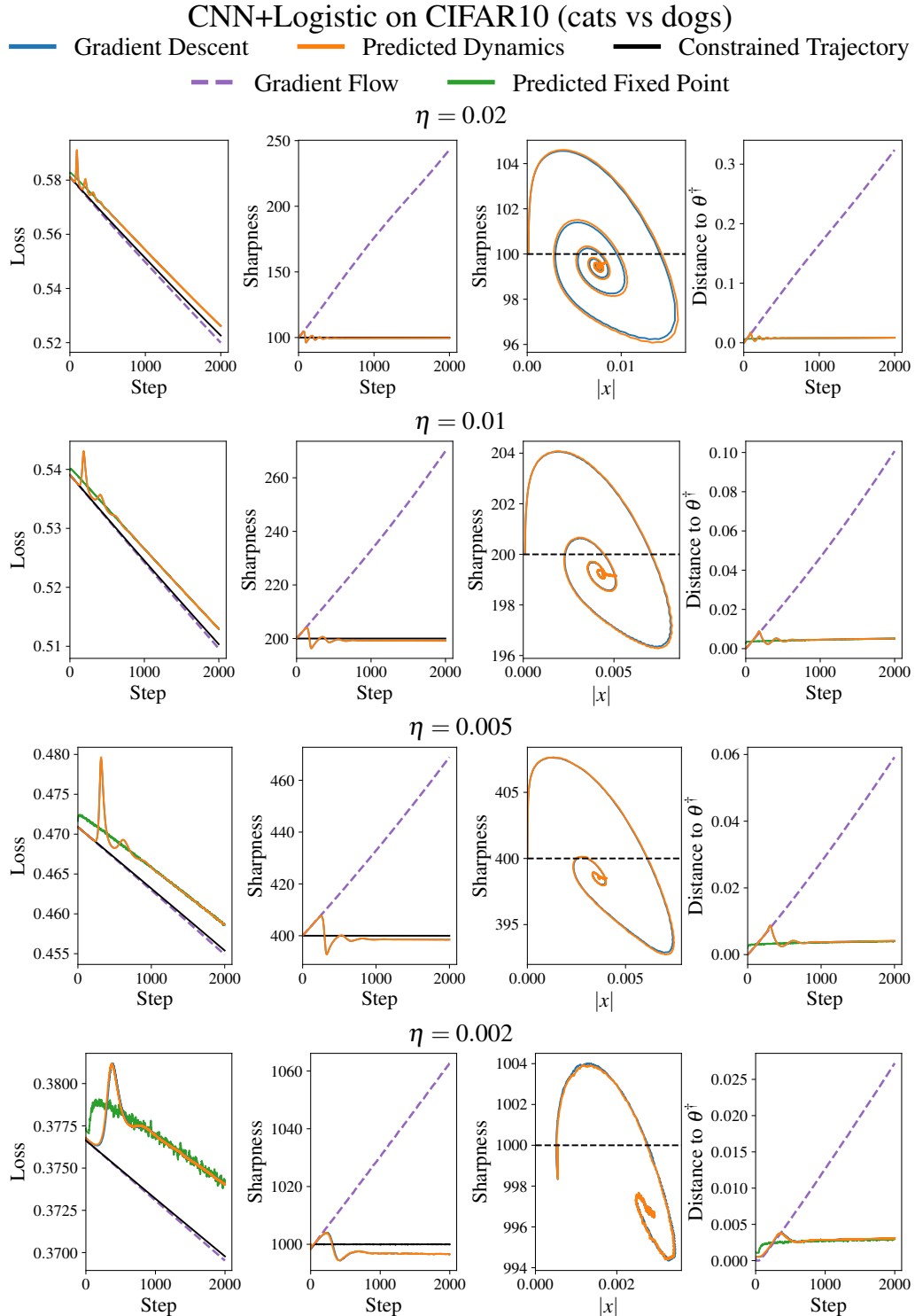

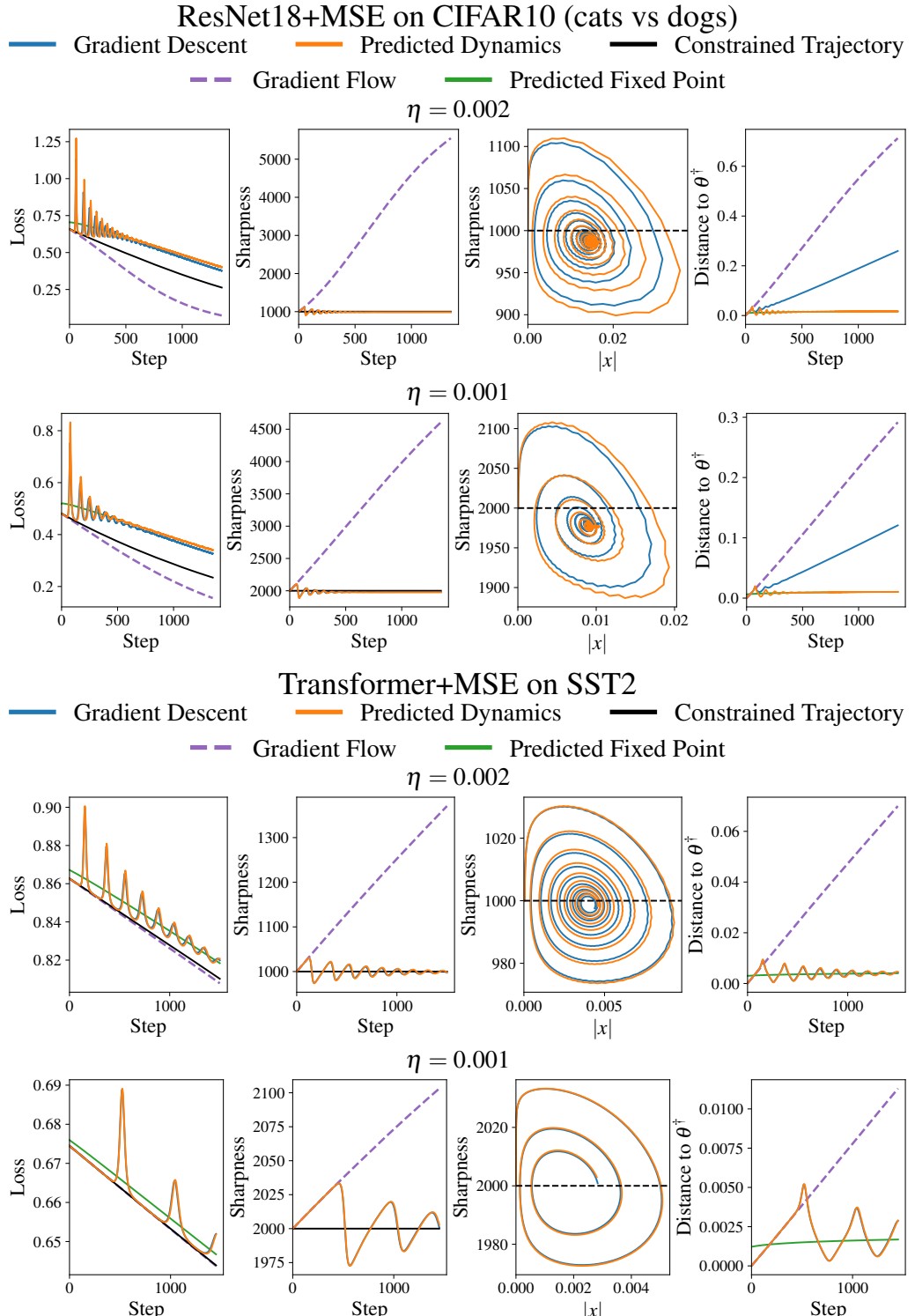

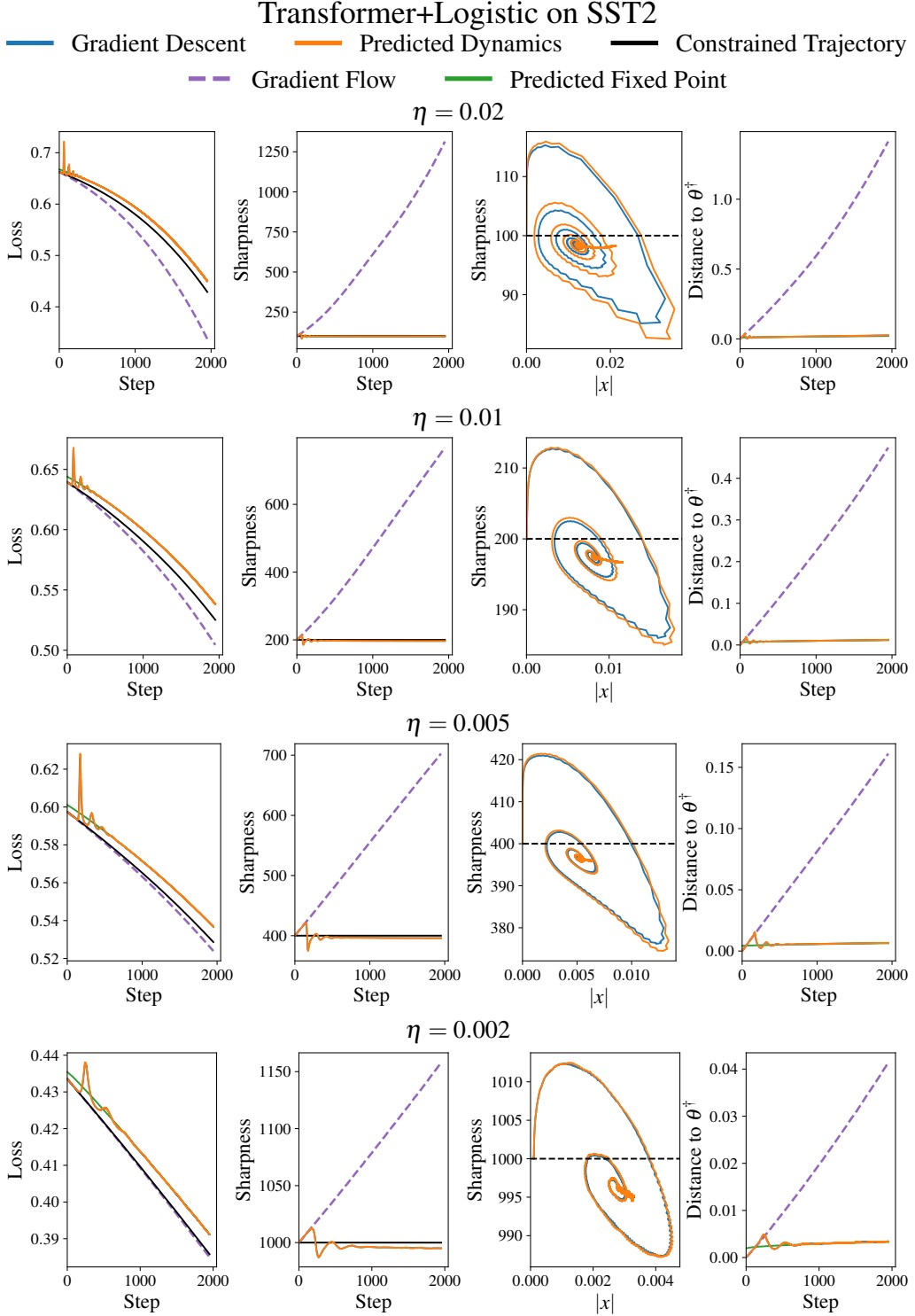

## H  ADDITIONAL DISCUSSION

**A Precise Criterion for Self-Stabilization**   Our theoretical results in Sections 4 and 5 give suf-
ficient conditions for self-stabilization as evidenced by Theorem 1. However, these assumptions
may not be strictly necessary and we believe that self-stabilization may hold under significantly
weaker assumptions. An important open question is understanding the precise conditions on the
loss function and learning rate that enable self-stabilization.

Furthermore, in Theorem 1 in Section 5, we give a quantitative bound on the time for which the
EOS trajectory and the constrained trajectory can be coupled. It is an interesting future direction
to understand whether coupling can be strengthened to hold for longer time periods to allow for
convergence to a KKT point of the constrained trajectory eq. (2).

**Multiple Unstable Eigenvalues**   Our work focuses on explaining edge of stability in the pres-
ence of a single unstable eigenvalue (Assumption 2). However, Cohen et al. (2021) observed that
progressive sharpening appears to apply to *all* eigenvalues, even after the largest eigenvalue has be-
come unstable. As a result, all of the top eigenvalues will successively enter edge of stability (see
Figure 5). In particular, Figure 5 shows that the dynamics are fairly well behaved in the period
when only a single eigenvalue is unstable, yet appear to be significantly more chaotic when multiple
eigenvalues are unstable.

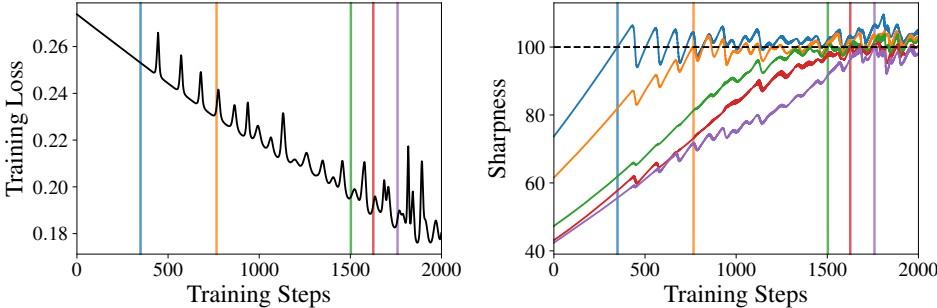

Figure 5: Edge of stability with multiple unstable eigenvalues. Each vertical line is the time at which
the corresponding eigenvalue of the same color becomes unstable.

One technical challenge with dealing with multiple eigenvalues is that, when the top eigenvalue
is not unique, the sharpness is no longer differentiable and it is unclear how to generalize our
analysis. However, one might expect that gradient descent can still be coupled to projected gra-
dient descent under the non-differentiable constraint $S(\theta_T^{\dagger}) \leq 2/\eta$. When there are $k$ unstable
eigenvalues, with corresponding eigenvectors $u_t^1, \ldots, u_t^k$, the constrained update is roughly equiv-
alent to projecting out the subspace $\text{span}\{\nabla^3 L_t(u_t^i, u_t^j) : i, j \in [k]\}$ from the gradient update
$-\eta \nabla L_t$. Demonstrating self-stabilization thus requires analyzing the dynamics in the subspace
$\text{span}\left( \{u_t^i : i \in [k]\} \cup \{\nabla^3 L_t(u_t^i, u_t^j) : i, j \in [k]\} \right)$. We leave investigating the dynamics of mul-
tiple unstable eigenvalues for future work.

**Connection to Sharpness Aware Minimization (SAM)**   Foret et al. (2021) introduced the
sharpness-aware minimization (SAM) algorithm, which aims to control sharpness by solving the
optimization problem $\min_\theta \max_{\|\delta\| \leq \epsilon} L(\theta + \delta)$. This is roughly equivalent to minimizing $S(\theta)$ over
all global minimizers, and thus SAM tries to explicitly minimize the sharpness. Our analysis shows
that gradient descent *implicitly* minimizes the sharpness, and for a fixed $\eta$ looks to minimize $L(\theta)$
subject to $S(\theta) = 2/\eta$.

**Connections to Warmup.**   Gilmer et al. (2021) demonstrated that *learning rate warmup*, which
consists of gradually increasing the learning rate, empirically leads to being able to train with a larger
learning rate. The self-stabilization property of gradient descent provides a plausible explanation for
this phenomenon. If too large of an initial learning rate $\eta_0$ is chosen (so that $S(\theta_0)$ is much greater

than $2/\eta_0$), then the iterates may diverge before self stabilization can decrease the sharpness to $2/\eta_0$. On the other hand, if the learning rate is chosen that $S(\theta_0)$ is only slightly greater than $2/\eta_0$, self-stabilization will decrease the sharpness to $2/\eta_0$. Repeatedly increasing the learning rate slightly could then lead to small decreases in sharpness without the iterates diverging, thus allowing training to proceed with a large learning rate.

**Connection to Weight Decay and Sharpness Reduction.** Lyu et al. (2022) proved that when the loss function is scale-invariant, gradient descent with weight decay and sufficiently small learning rate converges leads to reduction of the *normalized* sharpness $S(\theta/\|\theta\|)$. In fact, the mechanism behind the sharpness reduction is exactly the self-stabilization force described in this paper restricted to the setting in (Lyu et al., 2022). We present here a heuristic derivation of this equivalence.

First, we show that *any* scale invariant satisfies our assumptions when trained with weight decay.

**Lemma 6.** *Let $f$ be a scale invariant loss function, i.e. $f(\theta) = f(c\theta)$. Let $L(\theta) = f(\theta) + \frac{\lambda}{2}\|\theta\|^2$. Then for any local minimizer $\theta$ of $f(\theta)$ such that $S(\theta) = 2/\eta$,*

- $\nabla L(\theta) \perp u(\theta)$

- $\rho_4 = O(\eta \rho_3^2)$

- $\alpha(\theta) > 0$

- $\frac{\alpha(\theta)}{\|\nabla L(\theta)\|\|\nabla S(\theta)\|} = \Theta(1)$

- $\|\nabla S(\theta)\| = \Theta(\rho_3)$

Our primary result is that gradient descent solves the constrained problem $\min_\theta L(\theta)$ such that $S(\theta) \leq 2/\eta$. Let $S_f(\theta)$ denote the largest eigenvalue of $\nabla^2 f(\theta)$. To prove equivalence to the sharpness reduction, we will need the following lemma from (Lyu et al., 2022) which follows from the scale invariance of the $f$:

$$S_f(\theta) = \frac{1}{\|\theta\|^2} S_f(\theta/\|\theta\|).$$

Let $\overline{\theta} := \frac{\theta}{\|\theta\|}$. Then we have the following equality between minimization problems:

$$\min_\theta L(\theta) \quad \text{such that} \quad S(\theta) \leq 2/\eta$$

$$\iff \min_\theta f(\overline{\theta}) + \lambda \frac{\|\theta\|^2}{2} \quad \text{such that} \quad S_f(\theta) \leq 2/\eta - \lambda$$

$$\iff \min_{\overline{\theta}, \|\theta\|} f(\overline{\theta}) + \lambda \frac{\|\theta\|^2}{2} \quad \text{such that} \quad \frac{1}{\|\theta\|^2} S_f(\overline{\theta}) \leq \frac{2 - \eta\lambda}{\eta}$$

$$\iff \min_{\overline{\theta}} f(\overline{\theta}) + \frac{\eta\lambda}{2 - \eta\lambda} S_f(\overline{\theta})$$

where the last line follows from the scale-invariance of the loss function. In particular if $\eta\lambda$ is sufficiently small and the dynamics are initialized near a global minimizer of the loss, this will converge to the solution of the constrained problem:

$$\min_{\|\overline{\theta}\|=1} S_f(\overline{\theta}) \quad \text{such that} \quad f(\overline{\theta}) = 0.$$

### H.1 SCALE INVARIANT LEMMAS

Let $\theta$ denote an arbitrary parameter and let $\overline{\theta} = \theta/\|\theta\|$. Throughout this section, let $f$ be a scale invariant function with non-vanishing Hessian.

**Lemma 7.**

$$\nabla f(\theta) = \frac{P_\theta^\perp \nabla f(\overline{\theta})}{\|\theta\|}$$

$$\nabla^2 f(\theta) = \frac{P_\theta^\perp \nabla^2 f(\overline{\theta}) P_\theta^\perp - P_\theta^\perp \nabla f(\overline{\theta})\overline{\theta}^T - \overline{\theta}(P_\theta^\perp \nabla f(\overline{\theta}))^T}{\|\theta\|^2}$$

*Proof.* We start with the equality:

$$f(\theta) = f(\overline{\theta}).$$

Differentiating with respect to $\theta$ and using that $\nabla_\theta \overline{\theta} = \frac{P_{\overline{\theta}}^\perp}{\|\theta\|}$ gives,

$$\nabla f(\theta) = \frac{P_\theta^\perp \nabla f(\overline{\theta})}{\|\theta\|}.$$

Differentiating this again gives:

$$\nabla^2 f(\theta) = \frac{P_\theta^\perp \nabla^2 f(\overline{\theta}) P_\theta^\perp - P_\theta^\perp \nabla f(\overline{\theta})\overline{\theta}^T - \overline{\theta}(P_\theta^\perp \nabla f(\overline{\theta}))^T}{\|\theta\|^2}.$$

$\square$

A few corollaries immediately follow:

**Corollary 2.** *For any critical point $\theta$ of $f$,*

$$\nabla^2 f(\theta) = \frac{P_\theta^\perp \nabla^2 f(\overline{\theta}) P_\theta^\perp}{\|\theta\|^2}.$$

**Corollary 3.** *For any critical point $\theta$ of $f$,*

$$u(\theta) \perp \nabla L(\theta)$$

*Proof.* Note that from Corollary 2, the top eigenvector of $\nabla^2 f$ is perpendicular to $\theta$. In addition,

$$\nabla^2 L(\theta) = \nabla^2 f(\theta) + \lambda I$$

so this is also the top eigenvector of $\nabla^2 L(\theta)$, i.e. $u(\theta)$. Finally,

$$\nabla L(\theta) = \nabla f(\theta) + \lambda\theta = \lambda\theta$$

which is parallel to $\theta$ and concludes the proof. $\square$

**Lemma 8.**

$$\nabla S(\theta) = \frac{P_{\overline{\theta}}^\perp \nabla S(\overline{\theta}) - (S(\overline{\theta}) - \lambda)\overline{\theta}}{\|\theta\|^3}$$

*Proof.* Let $S_f(\theta)$ denote the largest eigenvalue of $\nabla^2 f(\theta)$. Then by scale invariance, $\nabla^2 f(\theta) = \nabla^2 f(\overline{\theta})/\|\theta\|^2$. This implies that

$$S_f(\theta) = \frac{S_f(\overline{\theta})}{\|\theta\|^2}.$$

Differentiating this gives:

$$\nabla S_f(\theta) = \frac{P_{\overline{\theta}}^\perp \nabla S_f(\overline{\theta}) - S_f(\overline{\theta})\overline{\theta}}{\|\theta\|^3}.$$

Finally, we have from $\nabla^2 L(\theta) = \nabla^2 f(\theta) + \lambda I$ that $S(\theta) = S_f(\theta) + \lambda$ so

$$\nabla S(\theta) = \frac{P_{\overline{\theta}}^\perp \nabla S(\overline{\theta}) - (S(\overline{\theta}) - \lambda)\overline{\theta}}{\|\theta\|^3}$$

$\square$

**Lemma 9.** *Let*

$$\rho_3 = \sup_{S(\theta) \le 4/\eta} \left\| \nabla^3 L(\theta) \right\|_{op} \quad and \quad \rho_4 = \sup_{S(\theta) \le 4/\eta} \left\| \nabla^4 L(\theta) \right\|_{op}.$$

*Then*

$$\rho_3 = \Theta(\eta^{-3/2}) \quad and \quad \rho_4 = O(\eta^{-2}).$$

*In particular, $\rho_4 = O(\eta \rho_3^2)$.*

*Proof.* Note that $S(\theta) < 4/\eta$ implies that

$$4/\eta = \frac{S_f(\bar{\theta})}{\|\theta\|^2} + \lambda \implies \|\theta\| = \Theta \eta^{-1/2}.$$

Therefore,

$$\nabla^3 L(\theta) = \frac{\nabla^3 L(\bar{\theta})}{\|\theta\|^3} = O(\theta^{-3/2})$$

and

$$\nabla^4 L(\theta) = \frac{\nabla^4 L(\bar{\theta})}{\|\theta\|^4} = O(\theta^{-2})$$

by compactness of $S^{d-1}$. In addition, note that $\left\| \nabla^3 L(u, u) \right\| = \left\| \nabla S(\theta) \right\| \ge \frac{S(\bar{\theta}) - \lambda}{\|\theta\|^3} = \Theta(\eta^{-3/2})$
so $\rho_3 = \Theta(\eta^{-3/2})$. $\qquad \square$

**Lemma 10.** *At any second order stationary point $\theta$ of $f$,*

$$\alpha(\theta) = \frac{\lambda(S(\bar{\theta}) - \lambda)}{\|\theta\|^2} > 0.$$

*Proof.*

$$\alpha(\theta) = -\nabla L(\theta) \cdot \nabla S(\theta) = -\lambda \theta \cdot \left[ -\frac{(S(\bar{\theta}) - \lambda)\bar{\theta}}{\|\theta\|^3} \right] = \frac{\lambda(S(\bar{\theta}) - \lambda)}{\|\theta\|^2}.$$

$\qquad \square$

**Lemma 11.** *At any second order stationary point $\theta$ of $f$,*

$$\frac{\alpha(\theta)}{\|\nabla L(\theta)\| \|\nabla S(\theta)\|} = \frac{\lambda(S(\bar{\theta}) - \lambda)}{\|\theta\|^2} = \Theta(1).$$

*Proof.*

$$\frac{\alpha(\theta)}{\|\nabla L(\theta)\| \|\nabla S(\theta)\|} = \frac{\lambda(S(\bar{\theta}) - \lambda)}{\|\theta\|^2} \cdot \frac{1}{\lambda \|\theta\|} \cdot \frac{\|\theta\|^3}{\sqrt{\left\| P_{\bar{\theta}}^{\perp} \nabla S(\bar{\theta}) \right\|^2 + (S(\bar{\theta}) - \lambda)^2}}$$

$$= \frac{1}{\sqrt{1 + \left\| \frac{P_{\bar{\theta}}^{\perp} \nabla S(\bar{\theta})}{S(\bar{\theta}) - \lambda} \right\|^2}}$$

$$= \Theta(1)$$

where the last step follows from compactness of $S^{d-1}$ and the fact that $\nabla^2 f$ is non-vanishing. $\qquad \square$

*Proof of Lemma 6.* The lemma is simply a restatement of Corollary 3, Lemma 10, Lemma 9, and Lemma 11. $\qquad \square$

## I PROOFS

### I.1 PROPERTIES OF THE CONSTRAINED TRAJECTORY

We next prove several nice properties of the constrained trajectory. First, we require the following auxiliary lemma, which shows that several quantities are Lipschitz in a neighborhood around the constrained trajectory:

**Definition 8** (Lipschitz Sets). $\mathcal{S}_t := B(\theta_t^\dagger, \frac{2-c}{4\eta\rho_3})$ where $c$ is the constant in Assumption 2 and $B(x, r)$ denotes the ball of radius $r$ centered at $x$.

**Lemma 12** (Lipschitz Properties).

1. $\theta \to \nabla L(\theta)$ is $O(\eta^{-1})$-Lipschitz in each set $\mathcal{S}_t$.

2. $\theta \to \nabla^2 L(\theta)$ is $\rho_3$-Lipschitz with respect to $\|\cdot\|_2$.

3. $\theta \to \lambda_i(\nabla^2 L(\theta))$ is $\rho_3$-Lipschitz.

4. $\theta \to u(\theta)$ is $O(\eta\rho_3)$-Lipschitz in each set $\mathcal{S}_t$.

5. $\theta \to \nabla S(\theta)$ is $O(\eta\rho_3^2)$-Lipschitz in each set $\mathcal{S}_t$.

*Proof.* The Lipschitzness of $\nabla^2 L(\theta)$ follows immediately from the bound $\left\|\nabla^3 L(\theta)\right\|_{op} \le \rho_3$. Weil's inequality then immediately implies the desired bound on the Lipschitz constant of the eigenvalues of $\nabla^2 L(\theta)$. Therefore for any $t$, we have for all $\theta \in \mathcal{S}_t$:

$$\lambda_1(\nabla^2 L(\theta)) - \lambda_2(\nabla^2 L(\theta)) \ge \lambda_1(\nabla^2 L(\theta)) - \lambda_2(\nabla^2 L(\theta)) - 2\rho_3 \frac{2-c}{4\eta\rho_3} \ge \frac{2-c}{2\eta}.$$

Next, from the derivative of eigenvector formula:

$$\begin{aligned}
\|\nabla u(\theta)\|_2 &= \left\|(\lambda_1(\nabla^2 L(\theta))I - \nabla^2 L(\theta))^\dagger \nabla^3 L(\theta)(u(\theta))\right\|_2 \\
&\le \frac{\rho_3}{\lambda_1(\nabla^2 L(\theta)) - \lambda_2(\nabla^2 L(\theta))} \\
&\le \frac{2\eta\rho_3}{2-c} \\
&= O(\eta\rho_3)
\end{aligned}$$

which implies the bound on the Lipschitz constant of $u$ restricted to $\mathcal{S}_t$. Finally, because $\nabla S(\theta) = \nabla^3 L(\theta)(u(\theta), u(\theta))$,

$$\left\|\nabla^2 S(\theta)\right\|_2 \le \left\|\nabla^4 L(\theta)\right\|_{op} + 2\left\|\nabla^3 L(\theta)\right\|_{op}\|\nabla u(\theta)\|_2 \le O(\rho_4 + \eta\rho_3^2) \le O(\eta\rho_3^2)$$

where the second to last inequality follows from the bound on $\|\nabla u(\theta)\|_2$ restricted to $\mathcal{S}_t$ and the last inequality follows from Assumption 3. $\square$

**Lemma 13** (First-order approximation of the constrained trajectory update $\{\theta_t^\dagger\}$). *For all $t \le \mathcal{T}$,*
$$\theta_{t+1}^\dagger = \theta_t^\dagger - \eta P_{u_t, \nabla S_t}^\perp \nabla L_t + O(\epsilon^2 \cdot \eta\|\nabla L_t\|) \quad and \quad S_t = 2/\eta.$$

*Proof.* We will prove by induction that $S_t = 2/\eta$ for all $t$. The base case follows from the definitions of $\theta_0, \theta_0^\dagger$. Next, assume $S(\theta_t^\dagger) = 0$ for some $t \ge 0$. Let $\theta' = \theta_t^\dagger - \eta\nabla L_t$. Then because $\theta_t^\dagger \in \mathcal{M}$ we have $\left\|\theta_{t+1}^\dagger - \theta'\right\| \le \left\|\theta_t^\dagger - \theta'\right\| = \eta\|\nabla L_t\|$. Then because $\theta_{t+1}^\dagger = \text{proj}_\mathcal{M}(\theta')$, the KKT conditions for this minimization problem imply that there exist $x, y$ with $y \ge 0$ such that

$$\begin{aligned}
\theta_{t+1}^\dagger &= \theta_t^\dagger - \eta\nabla L_t - x\nabla_\theta[\nabla L(\theta) \cdot u(\theta)]\Big|_{\theta=\theta_{t+1}^\dagger} - y\nabla S_{t+1} \\
&= \theta_t^\dagger - \eta\nabla L_t - x\left[S_{t+1}u_{t+1} + \nabla u_{t+1}^T \nabla L_{t+1}\right] - y\nabla S_{t+1} \\
&= \theta_t^\dagger - \eta\nabla L_t - x[S_{t+1}u_{t+1} + O(\eta\rho_3\|\nabla L_{t+1}\|)] - y\nabla S_{t+1} \\
&= \theta_t^\dagger - \eta\nabla L_t - x[S_t u_t + O(\eta\rho_3\|\nabla L_t\|)] - y\left[\nabla S_t + O(\eta^2\rho_3^2\|\nabla L_t\|)\right] \\
&= \theta_t^\dagger - \eta\nabla L_t - xS_t u_t - y\nabla S_t + O\left((|x|\eta\rho_3 + |y|\eta^2\rho_3^2)\|\nabla L_t\|\right).
\end{aligned}$$

Next, note that we can decompose $\nabla S_t = u_t(\nabla S_t \cdot u_t) + \nabla S_t^\perp$:

$$\theta_{t+1}^\dagger = \theta_t^\dagger - \eta\nabla L_t - [xS_t + y(\nabla S_t \cdot u_t)]u_t - y\nabla S_t^\perp + O\big((|x|\eta\rho_3 + |y|\eta^2\rho_3^2)\|\nabla L_t\|\big).$$

Let $s_t = \frac{\nabla S_t^\perp}{\|\nabla S_t^\perp\|}$. We can now perform the change of variables

$$(x', y') = \big(xS_t + y(\nabla S_t \cdot u_t), y\|\nabla S_t^\perp\|\big), \quad (x, y) = \left(\frac{x' - y'\frac{\nabla S_t \cdot u_t}{\|\nabla S_t^\perp\|}}{S_t}, \frac{y'}{\|\nabla S_t^\perp\|}\right)$$

to get

$$\theta_{t+1}^\dagger = \theta_t^\dagger - \eta\nabla L_t - x'u_t - y's_t + O\big(\eta^2\rho_3\|\nabla L\|(|x'| + |y'|)\big).$$

Note that

$$O(\eta^2\rho_3\|\nabla L\|(|x| + |y|)) \leq \frac{\sqrt{x^2 + y^2}}{2} \tag{12}$$

for sufficiently small $\epsilon$ so because $\left\|\theta_{t+1}^\dagger - \theta'\right\| \leq \eta\|\nabla L_t\|$ we have

$$\frac{\sqrt{x^2 + y^2}}{2} \leq \left\|\theta_{t+1}^\dagger - \theta'\right\| \leq \eta\|\nabla L_t\|$$

so $x, y = O(\eta\|\nabla L_t\|)$. Therefore,

$$\theta_{t+1}^\dagger = \theta_t^\dagger - \eta\nabla L_t - x'u_t - y's_t + O\big(\eta^3\rho_3\|\nabla L\|^2\big)$$
$$= \theta_t^\dagger - \eta\nabla L_t - x'u_t - y's_t + O\big(\epsilon^2 \cdot \eta\|\nabla L_t\|\big)$$

Then Taylor expanding $\nabla L_{t+1}$ around $\theta_t^\dagger$ gives

$$\nabla L_{t+1} \cdot u_{t+1} = \nabla L_t \cdot u_t + (\nabla L_{t+1} - \nabla L_t) \cdot u_t + \nabla L_{t+1} \cdot (u_{t+1} - u_t)$$
$$= u_t^T \nabla^2 L_t\big[-\eta\nabla L_t - x'u_t - y's_t + O\big(\epsilon^2 \cdot \eta\|\nabla L_t\|\big)\big] + O\big(\epsilon^2 \cdot \|\nabla L_t\|\big)$$
$$= -x'S_t + O\big(\epsilon^2 \cdot \|\nabla L_t\|\big)$$

so $x' = O(\epsilon^2 \cdot \eta\|\nabla L_t\|)$. We can also Taylor expand $S_{t+1}$ around $\theta_t^\dagger$ and use that $S_t = 2/\eta$ to get

$$S_{t+1} = 2/\eta + \nabla S_t \cdot \left[-\eta\nabla L_t - x'u_t - y's_t + O\big(\eta^3\rho_3\|\nabla L_t\|^2\big)\right] + O\big(\epsilon^2 \cdot \rho_3\eta\|\nabla L_t\|\big)$$
$$= 2/\eta + \eta\alpha_t - y'\|\nabla S_t^\perp\| + O\big(\epsilon^2 \cdot \rho_3\eta\|\nabla L_t\|\big).$$

Now note that for $\epsilon$ sufficiently small we have

$$O\big(\epsilon^2 \cdot \rho_3\eta\|\nabla L_t\|\big) \leq O\big(\epsilon^2 \cdot \eta\alpha_t\big) \leq \eta\alpha_t.$$

Therefore if $y' = 0$, we would have $S_{t+1} > 2/\eta$ which contradicts $\theta_{t+1}^\dagger \in \mathcal{M}$. Therefore $y' > 0$ and therefore $y > 0$, which by complementary slackness implies $S_{t+1} = 2/\eta$. This then implies that

$$-\eta\nabla L_t \cdot \nabla S_t^\perp - y'\|\nabla S_t^\perp\| + O\big(\epsilon^2 \cdot \rho_3\eta\|\nabla L_t\|\big) = 0 \implies y' = -\eta\nabla L_t \cdot \frac{\nabla S_t^\perp}{\|\nabla S_t^\perp\|} + O\big(\epsilon^2 \cdot \eta\|\nabla L_t\|\big).$$

Putting it all together gives

$$\theta_{t+1}^\dagger = \theta_t^\dagger - \eta P_{\nabla S_t^\perp}^\perp \nabla L_t + O\big(\epsilon^2 \cdot \eta\|\nabla L_t\|\big)$$
$$= \theta_t^\dagger - \eta P_{u_t, \nabla S_t}^\perp \nabla L_t + O\big(\epsilon^2 \cdot \eta\|\nabla L_t\|\big)$$

where the last line follows from $u_t \cdot \nabla L_t = 0$. $\qquad\square$

**Lemma 14** (Descent Lemma for $\theta^\dagger$). *For all $t \leq \mathcal{T}$,*

$$L(\theta_{t+1}^\dagger) \leq L(\theta_t^\dagger) - \Omega\Big(\eta\big\|P_{u_t, \nabla S_t}^\perp \nabla L_t\big\|^2\Big).$$

*Proof.* Taylor expanding $L(\theta_{t+1}^\dagger)$ around $L(\theta_t^\dagger)$ and using Lemma 13 gives

$$L(\theta_{t+1}^\dagger) = L(\theta_t^\dagger) + \nabla L_t \cdot (\theta_{t+1}^\dagger - \theta_t^\dagger) + \frac{1}{2}(\theta_{t+1}^\dagger - \theta_t^\dagger)^T \nabla^2 L_t(\theta_{t+1}^\dagger - \theta_t^\dagger) + O\left(\rho_3 \left\|\theta_{t+1}^\dagger - \theta_t^\dagger\right\|^3\right)$$

$$= L(\theta_t^\dagger) - \eta\left\|P_{u_t,\nabla S_t}^\perp \nabla L_t\right\|^2 + \frac{\eta^2 \lambda_2(\nabla^2 L_t)\left\|P_{u_t,\nabla S_t}^\perp \nabla L_t\right\|^2}{2} + O\left(\eta^3 \rho_3 \|\nabla L_t\|^3\right)$$

$$= L(\theta_t^\dagger) - \frac{\eta(2-c)}{2}\left\|P_{u_t,\nabla S_t}^\perp \nabla L_t\right\|^2 + O\left(\eta^3 \rho_3 \|\nabla L_t\|^3\right).$$

Next, note that because $\gamma_t = \Theta(1)$ we have $\|\nabla L_t\| = O(\|P_{u_t,\nabla S_t}^\perp \nabla L_t\|)$. Therefore for $\epsilon$ sufficiently small,

$$O\left(\eta^3 \rho_3 \|\nabla L_t\|^3\right) = O(\epsilon^2 \cdot \eta \|\nabla L_t\|^2) \le \frac{\eta(2-c)}{4}\left\|P_{u_t,\nabla S_t}^\perp \nabla L_t\right\|^2.$$

Therefore,

$$L(\theta_{t+1}^\dagger) \le L(\theta_t^\dagger) - \frac{\eta(2-c)}{4}\left\|P_{u_t,\nabla S_t}^\perp \nabla L_t\right\|^2 = L(\theta_t^\dagger) - \Omega(\eta\left\|P_{u_t,\nabla S_t}^\perp \nabla L_t\right\|^2)$$

which completes the proof. $\square$

**Corollary 4.** *Let $L^\star = \min_\theta L(\theta)$. Then there exists $t \le \mathscr{T}$ such that*

$$\left\|P_{u_t,\nabla S_t}^\perp \nabla L_t\right\|^2 \le O\left(\frac{L(\theta_0^\dagger) - L^\star}{\eta \mathscr{T}}\right).$$

*Proof.* Inductively applying Lemma 14 we have that there exists an absolute constant $c$ such that

$$L^\star \le L(\theta_{\mathscr{T}}^\dagger) \le L(\theta_0^\dagger) - c\eta \sum_{t<\mathscr{T}} \left\|P_{u_t,\nabla S_t}^\perp \nabla L_t\right\|^2$$

which implies that

$$\min_{t<\mathscr{T}} \left\|P_{u_t,\nabla S_t}^\perp \nabla L_t\right\|^2 \le \frac{\sum_{t<\mathscr{T}} \left\|P_{u_t,\nabla S_t}^\perp \nabla L_t\right\|^2}{\mathscr{T}} \le O\left(\frac{L(\theta_0^\dagger) - L^\star}{\eta \mathscr{T}}\right).$$

$\square$

### I.2 PROOF OF THEOREM 1

We first require the following three lemmas, whose proofs are deferred to Appendix I.3.

**Lemma 15** (2-Step Lemma)**.** *Let*

$$r_t := v_{t+2} - \text{step}_{t+1}(\text{step}_t(v_t)).$$

*Assume that $\|v_t\| \le \epsilon^{-1}\delta$. Then*

$$\|r_t\| \le O\left(\epsilon^2 \delta \cdot \max\left(1, \frac{\|v_t\|}{\delta}\right)^3\right).$$

**Lemma 16.** *Assume that there exists constants $c_1, c_2$ such that for all $t \le \mathscr{T}$, $\|\mathring{v}_t\| \le c_2\delta$, $|\mathring{x}_t| \ge c_1\delta$. Then, for all $t \le \mathscr{T}$, we have*

$$\|v_t - \mathring{v}_t\| \le O(\epsilon\delta)$$

**Lemma 17.** *For $t \le \mathscr{T}$, $\|\mathring{v}_t\| \le O(\delta)$.*

With these lemmas in hand, we can prove Theorem 1.

*Proof of Theorem 1.* First, by Lemma 17, we have $\|\mathring{v}_t\| \leq O(\delta)$.

Next, by Lemma 16, we have

$$\theta_t - \theta_t^\dagger = v_t = \mathring{v}_t + O(\epsilon\delta).$$

Next, we Taylor expand to calculate $S(\theta_t)$:

$$
\begin{aligned}
S(\theta_t) &= S(\theta_t^\dagger) + \nabla S_t \cdot v_t + O(\eta\rho_3^2\|v_t\|^2) \\
&= 2/\eta + \nabla S_t^\perp \cdot v_t + \nabla S_t \cdot u_t u_t \cdot v_t + O(\eta\rho_3^2\delta^2) \\
&= 2/\eta + \nabla S_t^\perp \cdot \mathring{v}_t + \nabla S_t \cdot u_t u_t \cdot \mathring{v}_t + O(\rho_3\epsilon\delta + \eta\rho_3^2\delta^2) \\
&= 2/\eta + y_t + (\nabla S_t \cdot u_t)x_t + O(\eta^{-1}\epsilon^2).
\end{aligned}
$$

Finally, we Taylor expand the loss:

$$
\begin{aligned}
L(\theta_t) &= L(\theta_t^\dagger) + \nabla L_t \cdot v_t + \frac{1}{2}v_t^T \nabla^2 L_t v_t + O(\rho_3\|v_t\|^3) \\
&= L(\theta_t^\dagger) + \frac{1}{\eta}x_t^2 + \frac{1}{2}v_t^{\perp T}\nabla^2 L_t v_t^\perp + O(\rho_1\|v_t\| + \rho_3\|v_t\|^3) \\
&= L(\theta_t^\dagger) + \frac{1}{\eta}\mathring{x}_t^2 + \frac{1}{2}\mathring{v}_t^{\perp T}\nabla^2 L_t \mathring{v}_t^\perp + O(\eta^{-1}\delta^2\epsilon) \\
&= L(\theta_t^\dagger) + \frac{1}{\eta}\mathring{x}_t^2 + O(\eta^{-1}\delta^2\epsilon),
\end{aligned}
$$

where the last line follows from Assumption 5.

$\square$

### I.3 PROOF OF AUXILIARY LEMMAS

*Proof of Lemma 15.* Taylor expanding the update for $\theta_{t+1}$ about $\theta_t^\dagger$, we get

$$
\begin{aligned}
\theta_{t+1} &= \theta_t - \eta\nabla L(\theta_t) \\
&= \theta_t - \eta\nabla L_t - \eta\nabla^2 L_t v_t - \frac{1}{2}\eta\nabla^3 L_t(v_t, v_t) + O\left(\eta\rho_4\|v_t\|^3\right)
\end{aligned}
$$

Additionally, recall that the update for $\theta_{t+1}^\dagger$ is

$$\theta_{t+1}^\dagger = \theta_t^\dagger - \eta P_{\nabla S_t^\perp}^\perp \nabla L_t + O\left(\epsilon^2 \cdot \eta\|\nabla L_t\|\right).$$

Subtracting the previous 2 equations and expanding out $\nabla^3 L(v_t, v_t)$ via the non-worst-case bounds, we obtain

$$
\begin{aligned}
v_{t+1} &= (I - \eta \nabla^2 L_t)v_t - \eta(\nabla L_t - P^{\perp}_{\nabla S_t^{\perp}} \nabla L_t) - \frac{1}{2}\eta x_t^2 \nabla S_t - \eta x_t \nabla^3 L_t(u_t, v_t^{\perp}) - \frac{1}{2}\eta \nabla^3 L_t(v_t^{\perp}, v_t^{\perp}) \\
&\quad + O\left(\eta \rho_4 \|v_t\|^3 + \epsilon^2 \cdot \eta \|\nabla L_t\|\right) \\
&= (I - \eta \nabla^2 L_t)v_t - \eta\left[\frac{\nabla L \cdot \nabla S^{\perp}}{\|\nabla S^{\perp}\|^2}\right]\nabla S_t^{\perp} - \frac{1}{2}\eta x_t^2 \nabla S_t - \eta x_t \nabla^3 L_t(u_t, v_t^{\perp}) \\
&\quad + O\left(\eta \rho_3 \epsilon \|v_t\|^2 + \eta \rho_4 \|v_t\|^3 + \epsilon^2 \cdot \eta \|\nabla L_t\|\right) \\
&= (I - \eta \nabla^2 L_t)v_t + \eta \nabla S_t^{\perp}\left[\frac{\epsilon_t^2 - x_t^2}{2}\right] - \frac{1}{2}\eta x_t^2 \nabla S_t \cdot u_t u_t - \eta x_t \nabla^3 L_t(u_t, v_t^{\perp}) \\
&\quad + O\left(\epsilon^2 \cdot \frac{\|v_t\|^2}{\delta} + \epsilon^2 \cdot \frac{\|v_t\|^3}{\delta^2} + \epsilon^3 \delta\right) \\
&= (I - \eta \nabla^2 L_t)v_t + \eta \nabla S_t^{\perp}\left[\frac{\epsilon_t^2 - x_t^2}{2}\right] - \frac{1}{2}\eta x_t^2 \nabla S_t \cdot u_t u_t - \eta x_t \nabla^3 L_t(u_t, v_t^{\perp}) \\
&\quad + O\left(\epsilon^2 \delta \cdot \max\left(1, \frac{\|v_t\|}{\delta}\right)^3\right)
\end{aligned}
$$

We would first like to compute the magnitude of $v_{t+1}$.

$$
\|v_{t+1}\| = O\left(\|v_t\| + \eta \rho_3 \|v_t\|^2 + \eta \|\nabla L_t\| + \epsilon^2 \delta \cdot \max\left(1, \frac{\|v_t\|}{\delta}\right)^3\right).
$$

Observe that by definition of $\epsilon$ and $\delta$, and since $\|v_t\| \le \epsilon^{-1}\delta$

$$
O(\eta \rho_3 \|v_t\|^2) \le O\left(\|v_t\| \cdot \epsilon^{-1} \eta \rho_3 \delta\right) \le O\left(\|v_t\| \cdot \epsilon^{-1} \eta \sqrt{\rho_1 \rho_3}\right) \le O(\|v_t\|)
$$

$$
O\left(\epsilon^2 \delta \cdot \max\left(1, \frac{\|v_t\|}{\delta}\right)^3\right) \le O\left(\epsilon^2 \delta + \|v_t\| \cdot \epsilon^2 \cdot (\epsilon^{-1})^2\right) \le O\left(\epsilon^2 \delta + \|v_t\|\right).
$$

Hence

$$
\|v_{t+1}\| = O\left(\|v_t\| + \eta \|\nabla L_t\| + \epsilon^2 \delta\right) = O(\|v_t\| + \epsilon \delta).
$$

Note that we can bound

$$
\begin{aligned}
\|u_{t+1} - u_t\| \cdot \|v_{t+1}\| &= O\left(\eta^2 \rho_3 \|\nabla L_t\| \cdot (\|v_t\| + \epsilon \delta)\right) \\
&= O\left(\epsilon^2 \cdot (\|v_t\| + \epsilon \delta)\right) \\
&\le O\left(\epsilon^2 \cdot \max(\|v_t\|, \delta)\right).
\end{aligned}
$$

Therefore, the one-step update in the $u_t$ direction is:

$$
\begin{aligned}
x_{t+1} &= v_{t+1} \cdot u_{t+1} \\
&= v_{t+1} \cdot u_t + O\left(\epsilon^2 \cdot \max(\|v_t\|, \delta)\right) \\
&= -v_t \cdot u_t - \frac{1}{2}\eta x_t^2 \nabla S_t \cdot u_t - \eta x_t \nabla S_t \cdot v_t^{\perp} + O\left(\epsilon^2 \cdot \max(\|v_t\|, \delta) + \epsilon^2 \delta \cdot \max\left(1, \frac{\|v_t\|}{\delta}\right)^3\right) \\
&= -x_t(1 + \eta y_t) - \frac{1}{2}\eta x_t^2 \nabla S_t \cdot u_t + O\left(\epsilon^2 \delta \cdot \max\left(1, \frac{\|v_t\|}{\delta}\right)^3\right) \\
&= -x_t(1 + \eta y_t) - \frac{1}{2}\eta x_t^2 \nabla S_t \cdot u_t + O\left(\epsilon^2 \delta \cdot \max\left(1, \frac{\|v_t\|}{\delta}\right)^3\right) \\
&= -x_t(1 + \eta y_t) - \frac{1}{2}\eta x_t^2 \nabla S_t \cdot u_t + O(E_t),
\end{aligned}
$$

where we have defined the error term $E_t$ as

$$E_t := \epsilon^2 \delta \cdot \max\left(1, \frac{\|v_t\|}{\delta}\right)^3.$$

The update in the $v^\perp$ direction is

$$v_{t+1}^\perp = P_{u_{t+1}}^\perp \left[(I - \eta \nabla^2 L_t)v_t + \eta \nabla S_t^\perp \left[\frac{\epsilon_t^2 - x^2}{2}\right]\right] - \frac{1}{2}\eta x_t^2 \nabla S_t \cdot u_t P_{u_{t+1}}^\perp u_t - \eta x_t P_{u_{t+1}}^\perp \nabla^3 L_t(u_t, v_t^\perp)$$

$$+ O\left(\epsilon^2 \delta \cdot \max\left(1, \frac{\|v_t\|}{\delta}\right)^3\right)$$

$$= P_{u_{t+1}}^\perp \left[(I - \eta \nabla^2 L_t)P_{u_t}^\perp v_t + \eta \nabla S_t^\perp \left[\frac{\epsilon_t^2 - x^2}{2}\right]\right] - x_t P_{u_{t+1}}^\perp u_t - \frac{1}{2}\eta x_t^2 \nabla S_t \cdot u_t P_{u_{t+1}}^\perp u_t - \eta x_t P_{u_{t+1}}^\perp \nabla^3 L_t(u_t, v_t^\perp)$$

$$+ O\left(\epsilon^2 \delta \cdot \max\left(1, \frac{\|v_t\|}{\delta}\right)^3\right)$$

First, observe that

$$\left\|P_{u_{t+1}}^\perp u_t\right\| = \left\|u_t - u_{t+1}u_{t+1}^T u_t\right\| \leq \|u_t - u_{t+1}\|^2 \leq O(\|u_t - u_{t+1}\|)$$

Therefore we can control the first of the error terms as

$$\left\|x_t P_{u_{t+1}}^\perp u_t + \frac{1}{2}\eta x_t^2 \nabla S_t \cdot u_t P_{u_{t+1}}^\perp u_t\right\| \leq O\Big(\|u_t - u_{t+1}\| \cdot (\|v_t\| + \eta \rho_3 \|v_t\|^2)\Big)$$

$$\leq O(\|u_t - u_{t+1}\| \cdot \|v_t\|)$$

$$\leq O(\epsilon^2 \|v_t\|),$$

As for the second error term, we can decompose

$$\left\|\eta x_t P_{u_{t+1}}^\perp \nabla^3 L_t(u_t, v_t^\perp)\right\| \leq \eta \|v_t\| \Big(\left\|P_{u_t}^\perp \nabla^3 L_t(u_t, v_t^\perp)\right\| + \left\|P_{u_t}^\perp - P_{u_{t+1}}^\perp\right\| \left\|\nabla^3 L_t(u_t, v_t^\perp)\right\|\Big).$$

By Assumption 5, we have $\left\|P_{u_t}^\perp \nabla^3 L_t(u_t, v_t^\perp)\right\| \leq O(\epsilon \rho_3 \|v_t\|)$. Additionally, $\left\|P_{u_t}^\perp - P_{u_{t+1}}^\perp\right\| \leq O(\|u_t - u_{t+1}\|)$. Therefore

$$\left\|\eta x_t P_{u_{t+1}}^\perp \nabla^3 L_t(u_t, v_t^\perp)\right\| \leq O(\epsilon \rho_3 \|v_t\| \cdot \eta \|v_t\| + \eta \|v_t\| \|u_{t+1} - u_t\| \cdot \rho_3 \|v_t\|)$$

$$\leq O\Big(\epsilon \eta \rho_3 \|v_t\|^2 + \eta \rho_3 \|v_t\|^2 \epsilon^2\Big)$$

$$\leq O\left(\epsilon^2 \frac{\|v_t\|^2}{\delta} + \epsilon^2 \|v_t\|\right)$$

$$= O\left(\epsilon^2 \delta \cdot \max\left(1, \frac{\|v_t\|}{\delta}\right)^3\right)$$

where we used $\eta \rho_3 \|v_t\| = O(1)$. Altogether, we have

$$v_{t+1}^\perp = P_{u_{t+1}}^\perp \left[(I - \eta \nabla^2 L_t)P_{u_t}^\perp v_t + \eta \nabla S_t^\perp \left[\frac{\epsilon_t^2 - x^2}{2}\right]\right] + O\left(\epsilon^2 \delta \cdot \max\left(1, \frac{\|v_t\|}{\delta}\right)^3\right)$$

$$= P_{u_{t+1}}^\perp \left[(I - \eta \nabla^2 L_t)P_{u_t}^\perp v_t + \eta \nabla S_t^\perp \left[\frac{\epsilon_t^2 - x^2}{2}\right]\right] + O(E_t)$$

We next compute the two-step update for $x_t$:

$$x_{t+2} = -x_{t+1}(1 + \eta y_{t+1}) - \frac{1}{2}\eta x_{t+1}^2 \nabla S_{t+1} \cdot u_{t+1} + O(E_{t+1})$$

$$= x_t(1 + \eta y_t)(1 + \eta y_{t+1}) + \frac{\eta}{2}\Big(\eta y_t x_t^2 \nabla S_t \cdot u_t + x_t^2 \nabla S_t \cdot u_t - x_{t+1}^2 \nabla S_{t+1} \cdot u_{t+1}\Big) + O((1 + \eta \rho_3 \|v_t\|)E_t + E_{t+1}).$$

We previously obtained $\eta\rho_3\|v_t\| = O(1)$. Furthermore,

$$E_{t+1} = \epsilon^2\delta \cdot \max\left(1, \frac{\|v_{t+1}\|}{\delta}\right)^3$$

$$= O\left(\epsilon^2\delta \cdot \max\left(1, \frac{\|v_t\|}{\delta} + \epsilon\right)^3\right)$$

$$= O\left(\epsilon^2\delta \cdot \max\left(1, \frac{\|v_t\|}{\delta}\right)^3\right)$$

$$= O(E_t).$$

Hence

$$x_{t+2} = x_t(1+\eta y_t)(1+\eta y_{t+1}) + \frac{\eta}{2}\left(\eta y_t x_t^2\nabla S_t \cdot u_t + x_t^2\nabla S_t \cdot u_t - x_{t+1}^2\nabla S_{t+1}\cdot u_{t+1}\right) + O(E_t).$$

The first of these two error terms can be bounded as

$$\left|\frac{1}{2}\eta^2 y_t x_t^2\nabla S_t \cdot u_t\right| \le O\left(\eta^2\rho_3^2\|v_t\|^3\right) \le O\left(\epsilon^2 \cdot \frac{\|v_t\|^3}{\delta^2}\right).$$

As for the second term, we can bound

$$|\nabla S_{t+1}\cdot u_{t+1} - \nabla S_t \cdot u_t| \le |u_{t+1}\cdot(\nabla S_{t+1} - \nabla S_t)| + |\nabla S_t \cdot(u_{t+1} - u_t)|$$

$$\le \|\nabla S_{t+1} - \nabla S_t\| + O(\rho_3)\cdot\|u_{t+1} - u_t\|$$

$$\le O\left(\eta^2\rho_3^2\|\nabla L_t\|\right)$$

$$\le O(\epsilon^2\rho_3)$$

Additionally, we have

$$x_{t+1} = -x_t + O(\eta\rho_3\|v_t\|^2 + E_t).$$

Therefore

$$\eta\left|x_{t+1}^2\nabla S_{t+1}\cdot u_{t+1} - x_t^2\nabla S_t \cdot u_t\right| \le \eta x_t^2|\nabla S_{t+1}\cdot u_{t+1} - \nabla S_t \cdot u_t| + \eta(x_{t+1}^2 - x_t^2)|\nabla S_{t+1}\cdot u_{t+1}|$$

$$\le O\left(\eta\rho_3\|v_t\|^2 \cdot \epsilon^2 + \eta\rho_3\|v_t\|\left(\eta\rho_3\|v_t\|^2 + E_t\right)\right)$$

$$\le O\left(\epsilon^2\|v_t\| + \epsilon^2 \cdot \frac{\|v_t\|^3}{\delta^2} + E_t\right)$$

$$= O(E_t).$$

Altogether, the two-step update for $x_t$ is

$$x_{t+2} = x_t(1+\eta y_t)(1+\eta y_{t+1}) + O(E_t).$$

Additionally, the two-step update for $v_t^\perp$ is

$$v_{t+2}^\perp = P_{u_{t+2}}^\perp\left[(I - \eta\nabla^2 L_{t+1})P_{u_{t+1}}^\perp v_{t+1} + \eta\nabla S_{t+1}^\perp\left[\frac{\epsilon_{t+1}^2 - x_{t+1}^2}{2}\right]\right] + O(E_{t+1})$$

$$= P_{u_{t+2}}^\perp(I - \eta\nabla^2 L_{t+1})P_{u_{t+1}}^\perp(I - \eta\nabla^2 L_t)P_{u_t}^\perp v_t + \eta P_{u_{t+2}}^\perp(I - \eta\nabla^2 L_{t+1})P_{u_{t+1}}^\perp\nabla S_t^\perp\left[\frac{\epsilon_t^2 - x_t^2}{2}\right]$$

$$+ \eta P_{u_{t+2}}^\perp\nabla S_{t+1}^\perp\left[\frac{\epsilon_{t+1}^2 - x_{t+1}^2}{2}\right] + O(E_t).$$

Define $\bar{v}_{t+1} = \text{step}_t(v_t), \bar{v}_{t+2} = \text{step}_{t+1}(\bar{v}_t)$, and $\bar{x}_i = \bar{v}_i \cdot u_i, \bar{y}_i = \nabla S_i^\perp\cdot\bar{v}_i$ for $i \in \{t+1, t+2\}$. By the definition of step, one sees that

$$\left\|\bar{v}_{t+1}^\perp - v_{t+1}^\perp\right\| \le O(E_t).$$

and

$$|\overline{x}_{t+1} - x_{t+1}| \leq \frac{1}{2}\eta x_t^2 |\nabla S_t \cdot u_t| + O(E_t) \leq O(\eta\rho_3\|v_t\|^2 + E_t)$$

The update for $x$ after applying step is

$$\overline{x}_{t+2} = -\overline{x}_{t+1}(1 + \eta\overline{y}_{t+1})$$
$$= x_t(1 + \eta y_t)(1 + \eta\overline{y}_{t+1}).$$

Therefore

$$|x_{t+2} - \overline{x}_{t+2}| \leq O\big(|x_t|\eta\big|y_{t+1} - \overline{y}_{t+1}\big|\big) + O(E_t)$$
$$\leq O\big(\eta\rho_3\|v_t\|\big\|v_{t+1}^\perp - \overline{v}_{t+1}^\perp\big\|\big) + O(E_t)$$
$$\leq O(E_t).$$

Additionally, the update for $v^\perp$ is

$$\overline{v}_{t+2}^\perp = P_{u_{t+2}}^\perp(I - \eta\nabla^2 L_{t+1})P_{u_{t+1}}^\perp(I - \eta\nabla^2 L_t)P_{u_t}^\perp v_t + \eta P_{u_{t+2}}^\perp(I - \eta\nabla^2 L_{t+1})P_{u_{t+1}}^\perp \nabla S_t^\perp\left[\frac{\epsilon_t^2 - x_t^2}{2}\right]$$

$$+ \eta P_{u_{t+2}}^\perp \nabla S_{t+1}^\perp\left[\frac{\epsilon_{t+1}^2 - \overline{x}_{t+1}^2}{2}\right].$$

Therefore

$$\big\|v_{t+2}^\perp - \overline{v}_{t+2}^\perp\big\| \leq O\big(\eta\|\nabla S_{t+1}\|(x_{t+1}^2 - \overline{x}_{t+1}^2) + E_t\big)$$
$$\leq O(\eta\rho_3\|v_t\|\|\overline{x}_{t+1} - x_{t+1}| + E_t)$$
$$\leq O\big(\eta^2\rho_3^2\|v_t\|^3 + E_t\big)$$
$$\leq O\left(\epsilon^2 \cdot \frac{\|v_t\|^3}{\delta^2} + E_t\right)$$
$$= O(E_t)$$

Altogether, we get that

$$\|r_t\| \leq O(E_t) = O\left(\epsilon^2\delta \cdot \max\left(1, \frac{\|v_t\|}{\delta}\right)^3\right),$$

as desired. $\qquad\square$

*Proof of Lemma 16.* Define

$$w_t = \begin{cases} 0 & t \text{ if is even} \\ r_{t-1} & t \text{ if is odd} \end{cases}$$

and define the auxiliary trajectory $\widehat{v}$ by $\widehat{v}_0 = v_0$ and $\widehat{v}_{t+1} = \text{step}(\widehat{v}_t) + w_t$. I first claim that $\widehat{v}_t = v_t$ for all even $t \leq \mathscr{T}$, which we will prove by induction on $t$. The base case is given by assumption so assume the result for some even $t \geq 0$. Then,

$$v_{t+2} = \text{step}_{t+1}(\text{step}_t(v_t)) + r_t$$
$$= \text{step}_{t+1}(\text{step}_t(\widehat{v}_t)) + r_t$$
$$= \text{step}_{t+1}(\widehat{v}_{t+1}) + w_{t+1}$$
$$= \widehat{v}_{t+2}$$

which completes the induction.

Next, we will prove by induction that for $t \leq \mathscr{T}$,

$$\big\|\widehat{v}_t^\perp - \mathring{v}_t^\perp\big\|, |\widehat{x}_t - \mathring{x}_t| \leq O(\epsilon\delta) \leq c_2\delta.$$

By definition, $\widehat{v}_0 = v_0 = \mathring{v}_0$, so the claim is clearly true for $t = 0$. Next, assume the claim holds for $t$. If $t$ is even then $\|w_t\| = 0$; otherwise $\|v_t\| \leq 2c_2\delta$, and thus

$$\|w_t\| \leq O\left(\epsilon^2\delta \cdot \max{(1, c_2)}^3\right) \leq O(\epsilon^2\delta).$$

First observe that

$$\left\|\widehat{v}_{t+1}^{\perp} - \mathring{v}_{t+1}^{\perp}\right\| \leq \left\|(I - \eta\nabla^2 L_t)(\widehat{v}_t^{\perp} - \mathring{v}_t^{\perp})\right\| + \frac{\eta\rho_3\left|\widehat{x}_t^2 - \mathring{x}_t^2\right|}{2} + \|w_t\|$$

$$\leq \left(1 + \eta\left|\lambda_{min}(\nabla^2 L_t)\right|\right)\left\|\widehat{v}_t^{\perp} - \mathring{v}_t^{\perp}\right\| + O(\epsilon) \cdot |\widehat{x}_t - \mathring{x}_t| + O(\epsilon^2\delta)$$

$$\leq \left(1 + \eta\left|\lambda_{min}(\nabla^2 L_t)\right|\right)\left\|\widehat{v}_t^{\perp} - \mathring{v}_t^{\perp}\right\| + O(\epsilon\delta) \cdot \left|\frac{\widehat{x}_t - \mathring{x}_t}{\mathring{x}_t}\right| + O(\epsilon^2\delta)$$

Next, note that

$$\frac{\widehat{x}_{t+1}}{\mathring{x}_{t+1}} = \frac{(1 + \eta\widehat{y}_t)\widehat{x}_t + O(\epsilon^2\delta)}{(1 + \eta\mathring{y}_t)\mathring{x}_t + O(\epsilon^2\delta)}$$

$$= \frac{(1 + \eta\mathring{y}_t)\widehat{x}_t + O(\epsilon^2\delta) + O(\epsilon) \cdot \left\|\widehat{v}_t^{\perp} - \mathring{v}_t^{\perp}\right\|}{(1 + \eta\mathring{y}_t)\mathring{x}_t + O(\epsilon^2\delta)}$$

$$= \frac{\widehat{x}_t}{\mathring{x}_t} + O\left(\epsilon^2 + \frac{\epsilon}{\delta}\left\|\widehat{v}_t^{\perp} - \mathring{v}_t^{\perp}\right\|\right).$$

Therefore

$$\left|\frac{\widehat{x}_{t+1} - \mathring{x}_{t+1}}{\mathring{x}_{t+1}}\right| \leq \left|\frac{\widehat{x}_t - \mathring{x}_t}{\mathring{x}_t}\right| + O\left(\epsilon^2 + \frac{\epsilon}{\delta}\left\|\widehat{v}_t^{\perp} - \mathring{v}_t^{\perp}\right\|\right).$$

Let $d_t = \max\left(\left\|\widehat{v}_t^{\perp} - \mathring{v}_t^{\perp}\right\|, \delta\left|\frac{\widehat{x}_t - \mathring{x}_t}{\mathring{x}_t}\right|\right)$. Then

$$\left\|\widehat{v}_{t+1}^{\perp} - \mathring{v}_{t+1}^{\perp}\right\| \leq \left(1 + \eta\left|\lambda_{min}(\nabla^2 L_t)\right| + O(\epsilon)\right)d_t + O(\epsilon^2\delta)$$

$$\delta\left|\frac{\widehat{x}_{t+1} - \mathring{x}_{t+1}}{\mathring{x}_{t+1}}\right| \leq (1 + O(\epsilon))d_t + O(\epsilon^2\delta).$$

Therefore

$$d_{t+1} \leq \left(1 + \eta\left|\lambda_{min}(\nabla^2 L_t)\right| + O(\epsilon)\right)d_t + O(\epsilon^2\delta)$$

$$\leq (1 + O(\epsilon))d_t + O(\epsilon^2\delta),$$

so for $t \leq \mathscr{T}$ we have $d_{t+1} \leq O(\epsilon\delta)$. Therefore

$$\left\|\widehat{v}_{t+1}^{\perp} - \mathring{v}_{t+1}^{\perp}\right\|, |\widehat{x}_{t+1} - \mathring{x}_{t+1}| \leq O(\epsilon\delta) \leq c_2\delta,$$

so the induction is proven. Altogether, we get $\|\widehat{v}_t - \mathring{v}_t\| \leq O(\epsilon\delta)$ for all such $t$, as desired. □

*Proof of Lemma 17.* Recall that

$$x_{t+1}^* = -(1 + \eta y_t^*)x_t^* \quad \text{and} \quad y_{t+1}^* = \eta\sum_{s=0}^{t}\beta_{s\to t}\left[\frac{\delta_s^2 - x_s^{*2}}{2}\right],$$

Since $t \leq \frac{1}{\eta\max_t|\lambda_{min}(\nabla^2 L_t)|}$, we have that $\beta_{s\to t} = O(\rho_3^2)$, and thus

$$\mathring{y}_t \leq O(\rho_3^2)t\eta\delta^2 = O(\sqrt{\rho_1\rho_3}).$$

Therefore

$$|\mathring{x}_{t+1}| = (1 + \eta\mathring{y}_t)|\mathring{x}_t| \leq (1 + O(\epsilon))|\mathring{x}_t|.$$

Since $t \leq O(\epsilon^{-1})$, $|\mathring{x}_t|$ grows by at most a constant factor, and thus $|\mathring{x}_t| \leq O(\delta)$. Finally, recall that

$$\mathring{v}_{t+1}^{\perp} = \eta\sum_{s=0}^{t}P_{u_{t+1}}^{\perp}\left[\prod_{k=t}^{s+1}A_k\right]\nabla S_s^{\perp}\left[\frac{\delta_s^2 - x_s^{*2}}{2}\right].$$

By the triangle inequality,

$$\left\|\mathring{v}_{t+1}^{\perp}\right\| \leq O(\eta t\rho_3\delta^2) \leq O(\delta).$$

Therefore $\|\mathring{v}_t\| \leq O(\delta)$. □

