# OpenReview forum: "Self-Stabilization: The Implicit Bias of Gradient Descent at the Edge of Stability"
_ICLR.cc/2023/Conference — ICLR 2023 poster_

### Official Review · Reviewer_qpmy · 2022-10-24

**Confidence:** 3
**Correctness:** 3
**Technical Novelty And Significance:** 3
**Empirical Novelty And Significance:** 3
**Recommendation:** 8

**Clarity, Quality, Novelty And Reproducibility:**

The writing is good and clear. The idea of incorporating the cubic term is novel to me.

**Strength And Weaknesses:**

## Strength
- This work provides a new direction to understand the edge of stability problem. Though the author argues that their method can only explain the EoS phase. By the argument in Stage 1, I feel like it can also be used to explain the progressive sharpening phase. It would be very great if the author can help me to understand what's the gap to preclude them to claim that this method can be used to study the first phase.
- I really like the motivation section (Section 4), which is very clear and convincing to me.

## Weaknesses
- At the end of the first paragraph in Section 1.2, they say that "Unlike progressive sharpening ..., self stabilization is a general property of gradient descent." However, in Section 5, the main theorem relies on many assumptions, which are only verified on neural network training. Hence, I am a little bit concerned about its generality. Apart from the numerical verification, I can only find the toy example provided in Appendix B, which is just a cubic function. It would be great if the author can provide either more general examples, or verify that those assumptions hold for a general function class.
- As argued in Section 4.3, they show that the distance between $\theta_t$ and $\theta_t^{\dagger}$ is bounded by $\delta$. However, it is unclear how large this $\delta$ can be. Note that $\delta=\sqrt{2\alpha/\beta}$. In the appendix, it can be seen that $\alpha$ is not small. Hence, it is not clear to me why $\theta_t$ and $\theta_t^{\dagger}$ are close.

**Summary Of The Paper:**

This paper studies the edge of stability phenomenon. They propose to incorporate the cubic term in the convergence analysis and show that there is a self-stabilization property caused by this cubic term for general nonconvex optimization problems. Moreover, they show that GD is inherently regularizing the sharpness, i.e., it is close to a constraint optimization problem that explicitly adds the sharpness as the constraint. Last, they provide very promising simulations showing that the predicted trajectory mimics the actual GD trajectory on many realistic tasks.

**Summary Of The Review:**

Please see the above comments.

---

> ### Author Response · Authors · 2022-11-16
> **Response to Reviewer qpmy**
>
> We would like to thank the reviewer for your detailed and thoughtful review. Please let us know if you have any further questions or if anything is still unclear.
>
> > At the end of the first paragraph in Section 1.2, they say that "Unlike progressive sharpening ..., self stabilization is a general property of gradient descent." However, in Section 5, the main theorem relies on many assumptions, which are only verified on neural network training. Hence, I am a little bit concerned about its generality. Apart from the numerical verification, I can only find the toy example provided in Appendix B, which is just a cubic function. It would be great if the author can provide either more general examples, or verify that those assumptions hold for a general function class.
>
> This is an excellent point and we have updated the paper with a general function class which satisfies our assumptions. In particular, inspired by [1], we show that all scale invariant loss functions, e.g. resulting from batch normalization, satisfy our assumptions and we have added these findings to Appendix H.
>
> [1] Understanding the Generalization Benefit of Normalization Layers: Sharpness Reduction, Lyu et al. (2022)
>
> > “However, it is unclear how large this $\delta$ can be. Note that $\delta = \sqrt{2\alpha/\beta}$. In the appendix, it can be seen that $\alpha$ is not small. Hence, it is not clear to me why $\theta_t$ and $\theta^\dagger_t$ are close.
>
> While $\alpha$ is not particularly small, $\beta$ is incredibly large which leads to $\delta$ being very small. We did not include $\delta$ in Appendix E (verification of assumptions) as we never made any direct assumptions on $\delta$. However, you can see the empirical values for $\delta$ in Figure 3 and Appendix G.2. Because the fixed point is $(\delta,0)$, $\delta$ is represented by the green curve in the rightmost plots (fourth column). In particular $\delta \approx 0.002$ for the MLP, CNN, and Transformer in Figure 3 which is due to the large size of $\beta$.

---

### Official Review · Reviewer_dCeH · 2022-10-26

**Confidence:** 3
**Correctness:** 4
**Technical Novelty And Significance:** 3
**Empirical Novelty And Significance:** 2
**Recommendation:** 8

**Clarity, Quality, Novelty And Reproducibility:**

The paper is written clearly and I'd like to thank the authors for their great effort. There are a few small issues with the presentation, some of which I list below.

### Minor issues
I did not understand the meaning of this sentence: "Note that $y_t$ is approximately to the change in sharpness from $\theta^\star$ to $\theta^t$". Maybe "approximately equal"?
It's a bit strange that $^\star _{v_t}$ is defined in Appendix C and not explained, at least on some basic level, in Section 5.2. Since $^\star _{x_t}$ and $^\star _{t_t}$ depend on $^\star _{v_t}$, it makes it a bit harder to understand their meaning.
I did not understand why it is fine to assume that $\min{t\le \mathcal{J}}|^\star _{x_t}| \ge c_1\delta$.


**Strength And Weaknesses:**

## Strengths
1. The work sheds some light on the recently observed phenomena in deep learning, which may help us understand how deep networks learn the data.
2. The proposed explanation is quite intuitive and is supported by empirical evaluations.
3. The paper has a good flow and starts with simple ideas that finally lead to the main result.

## Weaknesses
1. From the mathematical point of view, this paper is not strong. Assumption 4 intuitively implies self-stabilization because it guarantees a significant negative correlation between the gradients of loss and sharpness. While the conclusions still require quite a bit of technical work to do, which the authors did well, the assumptions are very stringent. I find this to be the main weakness of this paper.
2. The studied problem rarely comes with deterministic gradients, and the impact of noise in stochastic training is not considered.
3. The implications of the results do not seem to change our understanding of how neural networks need to be trained. The notion of self-stabilization does not seem to suggest much more than the edge of stability already tells us.

**Summary Of The Paper:**

The authors study a phenomenon they call "self-stabilization", which is proposed to explain common empirical observations such as sharpening and convergence at the edge of stability.

Formally, the authors consider the iterates of (discrete time) gradient descent and assume that the product between the gradient of the objective and the gradient of sharpness is negative, where sharpness is defined as the largest eigenvalue of the Hessian. For simplicity, the authors also assume that the Hessian has at most one negative eigenvalue. Under these assumptions, the authors present a simplified local analysis of the self-stabilization phenomenon. Then, they extend this analysis to general loss functions, but under much more restrictive assumptions, such as Assumption 4, on the gradients of sharpness and loss. The main result is given in Theorem 1, which shows how, up to high-order terms, the loss and sharpness coevolve.

Finally, the authors present numerical evaluations serving as a justification for their assumptions and validating some of their claims.

**Summary Of The Review:**

I enjoyed reading the paper and I believe it will be of interest to the community. While the assumptions behind the theory are far from perfect, the derivation is still of interest. The empirical evaluations are meaningful and support the theory.

---

> ### Author Response · Authors · 2022-11-16
> **Response to Reviewer dCeH (1/2)**
>
> We would like to thank the reviewer for your detailed and thoughtful review. Please let us know if you have any further questions or if anything is still unclear.
>
> > Assumption 4 intuitively implies self-stabilization because it guarantees a significant negative correlation between the gradients of loss and sharpness.
>
> While we agree that Assumption 4 is necessary, we do not believe that it “intuitively implies self-stabilization”. In particular, because the iterates move in the direction of the negative gradient, Assumption 4 implies that for gradient flow, $\frac{dS}{dt} = - \nabla L \cdot \nabla S > 0$ so the sharpness *increases* over time. We, however, show that the sharpness eventually begins to *decrease* again, which is a consequence of our four stage analysis in section 4.1.
>
> Also, as we mentioned to reviewer UEL4 in our response, we believe that assumptions 1,4 are best interpreted as focusing in on the interesting case $\alpha > 0$ which leads to edge of stability dynamics. The $\alpha < 0$ case is mostly trivial as gradient flow naturally decreases the sharpness so $S(\theta) < 2/\eta$ always holds and gradient descent $\approx$ gradient flow.
>
> > The studied problem rarely comes with deterministic gradients, and the impact of noise in stochastic training is not considered.
>
> We agree that this is a limitation of our current work and believe that this is among the most promising directions for future research. In particular, many analyses have identified implicit biases of SGD of the form “SGD on $L(\theta)$ $\approx$ gradient flow on $\tilde L(\theta)$” where $\tilde L(\theta)$ is a regularized loss. However, to the best of our knowledge, all such analyses assume stability, i.e. $S(\theta) \ll 2/\eta$. In [2], the authors conduct a series of experiments that demonstrate that there is some form of edge of stability behavior even for SGD, but that the limiting sharpness is strictly below $2/\eta$. We hope that our proof techniques and analysis can be used to understand the edge of stability dynamics in this more general setting as well.
>
> [2] Adaptive Gradient Methods at the Edge of Stability, Cohen et al. (2022)
>
> > The implications of the results do not seem to change our understanding of how neural networks need to be trained. The notion of self-stabilization does not seem to suggest much more than the edge of stability already tells us.
>
> We believe that the intuition behind self-stabilization as the mechanism driving the edge of stability dynamics carries with it lessons which are more broadly applicable and have the potential to influence neural network optimization in practice.
>
> For example, our story of self-stabilization demonstrates that far from being chaotic, as many people believed after first reading about the edge of stability phenomenon, the edge of stability dynamics are actually very controlled and can be well approximated by gradient flow on a constrained objective which is a well behaved process.
>
> Our analysis also uncovers how different model parameters affect the level of instability and the deviation from the constrained trajectory. For example, Figure 2 demonstrates that the size of the instability is strongly affected by the size $x_0$, which suggests that artificially adding noise to increase the size of $x_0$ should weaken the loss spikes. In fact, we observed exactly this behavior when running our experiments with varying precision. Experiments using 16 or 32 bit precision exhibited weaker loss spikes and increased stability compared to our experiments with 64 bit precision, precisely because the limited precision lower bounded how small $x_0$ could be.
>
> Finally, we believe that our work has strong implications for understanding the generalization effect of large learning rates which has been repeatedly observed in practice for both SGD and full batch gradient descent. Our work shows that for full batch gradient descent, this improved generalization is a result of running gradient flow on the trajectory $\min_\theta L(\theta)$ such that $S(\theta) \le 2/\eta$ which demonstrates that this generalization gap is due to this implicit sharpness constraint.

---

> > ### Comment · Reviewer_dCeH · 2022-11-21
> > **It's unclear why $\alpha$ must have a fixed sign**
> >
> > Thank you for your response. I still find Assumption 4 a very strong one. You're right that the case $\alpha<0$ is less interesting, but you do not mention that there is a third case where the sign of $\alpha$ can change. Unless I missed something, the question of why the scalar product has a particular sign is not answered, and it might be even more fundamental than the question of what happens once the sign of the product stabilizes.

---

> ### Author Response · Authors · 2022-11-16
> **Response to Reviewer dCeH (2/2)**
>
> > I did not understand the meaning of this sentence: "Note that $y_t$ is approximately equal to the change in sharpness from $\theta^\star$ to $\theta_t$.” Maybe “approximately equal”?
>
> Thank you for catching this typo, we have fixed this in the updated manuscript.
>
> > It's a bit strange that $v_t^\star$ is defined in Appendix C and not explained, at least on some basic level, in Section 5.2. Since $x_t^\star$ and $y_t^\star$ depend on $v_t$, it makes it a bit harder to understand their meaning.
>
> To be consistent with the analysis in Section 4, we focused on the projections of $v_t^\star$ onto $u_t$ and $\nabla S_t^\perp$. Lemma 4 demonstrates that these two one dimensional projections form a closed system as in Section 4. We could therefore have defined $x_t^\star,y_t^\star$ to be the solution to the eq (5). The full update for $v_t^\star$ in Appendix C is a result of a cubic Taylor expansion followed by eliminating all higher order terms which do not contribute to the dynamics. As the full update is somewhat complicated and technical, we decided to defer it to the appendix. However, we are open to any suggestions for ways to improve our presentation of the predicted dynamics in section 5.
>
> > I did not understand why it is fine to assume that $\min_{t \le \mathscr{T}} |x_t^\star| \ge c_1 \delta$.
>
> This is an empirical observation which is strictly necessary as when $x_t \approx 0$, the dynamics become unstable due to the dependence of the size of the fluctuations on $x_0$ (see Figure 2). This can also be ensured by adding small amounts of noise in the top eigenvector direction, which does empirically increase the stability of gradient descent.

---

### Official Review · Reviewer_UEL4 · 2022-10-26

**Confidence:** 3
**Correctness:** 4
**Technical Novelty And Significance:** 4
**Empirical Novelty And Significance:** Not applicable
**Recommendation:** 8

**Clarity, Quality, Novelty And Reproducibility:**

I'm not an expert in EoS, but the results seem novel and interesting to me.
The appendix contains various additional details and the proof of the main result (Theorem 1). I didn't check the proof, but the theorem looks plausible.

**Strength And Weaknesses:**

I quite like this paper. There are several other current studies of EoS (e.g., https://openreview.net/forum?id=p7EagBsMAEO, https://openreview.net/forum?id=R2M14I9LEwW), but the present paper paints an especially general and appealing picture of this effect. The exposition is very clear. The theory is confirmed by multiple experiments with real data.

One natural question that does not seem to have been answered in the paper is why can we generally expect Assumption 1, the key alignment assumption, to hold? It seems that the authors leave this question for future work?

**Summary Of The Paper:**

The paper studies the Edge-of-Stability effect from a rather general perspective. It identifies progressive sharpening and the oscillatory behavior of optimization trajectories at EoS as resulting simply from the alignment of the loss and sharpness gradients. This alignment is determined by a particular cubic term of the Taylor expansion of the loss. The resulting dynamics is related to the well-known Lotka-Volterra predator-prey model. The paper first discusses a simplified scenario where this effect can be easily demonstrated and understood. After that a theorem is stated addressing a general setting. Finally, a series of experiments with real world data sets and neural networks is presented, showing agreement with theory.

**Summary Of The Review:**

A good paper, accept.

---

> ### Author Response · Authors · 2022-11-16
> **Response to Reviewer UEL4**
>
> We would like to thank the reviewer for your detailed and thoughtful review. Please let us know if you have any further questions or if anything is still unclear.
>
> > One natural question that does not seem to have been answered in the paper is why can we generally expect Assumption 1, the key alignment assumption, to hold? It seems that the authors leave this question for future work?
>
> While we stated Assumption 1 (existence of progressive sharpening) as an assumption, this assumption is not strictly necessary. In particular, our analysis captures the gradient descent dynamics regardless of whether $\alpha > 0$ or $\alpha < 0$.
>
> When $\alpha < 0$, gradient flow naturally *decreases* the sharpness, and will forever remain in the regime where gradient descent $\approx$ gradient flow and $S(\theta) < 2/\eta$. The $\alpha < 0$ case is therefore somewhat trivial. The interesting case is $\alpha > 0$, which leads to the edge of stability dynamics observed in experiments and is the focus of this paper.
>
> We agree that the question of *why* progressive sharpening happens is still open; however, we believe that this question is orthogonal to the question of understanding the edge of stability dynamics.

---

### Decision · Program_Chairs · 2023-01-20

**Decision:**

Accept: poster

**Justification For Why Not Higher Score:**

While all reviews recommend accept, there were no advocates for a higher score.

**Justification For Why Not Lower Score:**

NA

**Metareview: Summary, Strengths And Weaknesses:**

The paper studies gradient descent at the edge of stability.

* Strengths are the clear exposition with intuitive and general explanations of observed phenomena in deep learning.
* Weaknesses are limitations to study noise in stochastic training and strong assumptions.

All reviewers concluded that this is a good paper that should be accepted. Some reservations were mentioned in regard to the mathematical strength and assumptions. The rebuttal clarified a few items from the initial reviews. Some of the limitations of the work were acknowledged by the authors in their responses. I agree that the article addresses an interesting topic and in view of the positive feedback from the reviewers I am recommending accept.

**Note From Pc:**

if the above contains the word "oral" or "spotlight" please see: "oral" presentation means -> notable-top-5% and "spotlight" means -> notable-top-25%. As stated in our emails, we are disassociating presentation type from AC recommendations

**Summary Of Ac-Reviewer Meeting:**

NA